# Assessment of SWAT spatial and temporal transferability for high altitude glacierised catchments

Maria Andrianaki[1], Juna Shrestha[1], Florian Kobierska[2], Nikolaos P. Nikolaidis[3], Stefano M. Bernasconi[1]

[1] Geological Institute, ETH Zurich, 8092 Zürich, Switzerland

[2] Agroscope, Reckenholzstrasse 191, CH-8046 Zürich

[3] Department of Environmental Engineering, Technical University of Crete, 73100 Chania, Greece

*Correspondence to*: Maria Andrianaki (mandrianaki@hotmail.com)

**Abstract.** In this study, we investigated the application and the transferability of the Soil Water and Assessment Tool (SWAT) in a partly glacierised alpine catchment, characterised by extreme climatic conditions and steep terrain. The model was initially calibrated for the 10 km$^2$ watershed of the Damma glacier Critical Zone Observatory (CZO) in central Switzerland using monitoring data for the period of 2009–2011 and then was validated with the measurements collected during 2012–2013 in the same area. Model performance was found to be satisfactory against both the Nash Sutcliffe criterion (NS) and a benchmark efficiency (BE). The transferability of the model was assessed by using the parameters calibrated on the small watershed and applying the model to the approximately 100 km$^2$ catchment that drains into the hydropower reservoir of the Göscheneralpsee and includes the Damma glacier CZO. Model results were compared to the reservoir inflow data from 1997 to 2010 and it was found that the model predicted successfully snowmelt timing and autumn recession but could not accurately capture the peak flow for certain years. Runoff was slightly overestimated from late May to June, when it is dominated by snowmelt, due to the fact that only one melt factor for both snowmelt and glacier melt was used. Finally, we investigated the response of the greater catchment to climate change using three different climate change scenarios and the results were compared to those of a previous study, where two different hydrological models, PREVAH and ALPINE 3D, were used. Predicted changes in future runoff and peak flow as well as seasonal dynamics are similar between the two studies. It is concluded that the methodology presented here, where SWAT is calibrated for a small watershed and then applied for a bigger area with similar climatic conditions and geographical characteristics, could work even under extreme conditions like ours. However, a greater attention should be given to the differences between glacial melt and snowmelt dynamics, since our findings indicate that the performance of the model as well as its transferability could be improved if different parameters for snowmelt and glacial melt were applied.

## 1 Introduction

The use of calibrated watershed models enables researchers and stakeholders to assess the impact of natural and management induced environmental changes and, as many studies have pointed out, is of high importance in water management (i.e. Arnold et al., 1998; Abbaspour et al., 2007). Climate change simulations provide crucial information for the assessment of its impact on water resources, water quality, and aquatic ecosystems (Farinotti et al., 2012; Aili et al., 2019). However, watershed modelling in high altitude alpine areas is rather challenging due to the rough terrain, heterogeneous land cover, extreme climatic conditions and glacier dynamics (Viviroli and Weingartner, 2004; Farinotti et al., 2012; Rahman et al., 2013), with the main challenge to be the lack of observed

and sufficient quality data in ungauged watersheds (Sivapalan et al., 2003; Viviroli et al., 2009b; Bocchiola et al.,
39  2011).


Modelling and predicting the runoff of ungauged watersheds is one of the big challenges that hydrologists face
today (Sivapalan et al., 2003; Hrachowitz et al., 2013). A common approach to address this problem is to calibrate
a hydrological model for a gauged watershed using observed data and then transfer the model to the ungauged
watershed by transferring the model parameters (Merz and Blöschl, 2003; Sivapalan et al., 2003). A great number
of methods have been suggested for transferring model parameters, which include regression techniques between
the model parameters and catchment attributes (e.g. Parajka et al., 2005; Deckers et al., 2010; Zhang et al., 2018)
and similarity approaches such as spatial proximity and physical similarity (e.g. Bárdossy, 2007; Wagener et al.,
2007; Patil and Stieglitz, 2014). However, as Thirel et al. (2015) point out, it is essential to asses and evaluate the
ability of the hydrological models to perform efficiently under conditions different from those in which they were
developed or calibrated.

The Soil and Water Assessment Tool (SWAT) developed by the USDA Agricultural Research Service (ARS) is a
public domain and open source integrated model and has been used worldwide for various applications. As a semi-
distributed model, it allows the spatial variation of the parameters by dividing the basin into a number of sub-
basins (Arnold et al., 1998; Srinivasan et al., 1998). It is equipped with a snowmelt algorithm based on a simple
temperature-index approach, which, although simple, is proved to be very effective in numerous studies (Hock,
2003) especially when net solar radiation is the dominant driving energy for snowmelt (Debele et al., 2010).

SWAT has been widely used in many studies for the simulation of runoff and nutrient cycling in agricultural and
forested sites. Although there is an increasing interest in applying SWAT on snow-dominated (Grusson et al.,
2015) and glacierised watersheds (Rahman et al., 2013; Garee et al., 2017; Omani et al., 2017), its transferability
under the extreme conditions of these high altitude environments has not been tested yet. In this study, we have a
quite unique situation of a small well gauged watershed, the Damma glacier watershed, which is part of the larger
catchment feeding the Göscheneralpsee reservoir, for which we have hydrological data thanks to its use by the
hydroelectric power plant. This way we were able to assess the transferability and upscaling of SWAT, by
calibrating the model for the Damma glacier watershed and then transferring it to the greater area feeding the
Göscheneralpsee reservoir. Subsequently, climate change simulations were conducted in order to assess the
transferability of the model on a temporal scale. The assessment was conducted by comparing our findings with
those of a previous study for the same area, which used two other hydrological models with different
characteristics, PREVAH and ALPINE3D (Kobierska et al., 2013).

**2 Study Site**
The Damma glacier watershed (Fig. 1a) is situated in the central Swiss Alps in Switzerland and was one of the
Critical Zone Observatories established within the European project SoilTrEC (Banwart et al., 2011). It is located
at an altitude between 1790 m and 3200 m above sea level, has a total area of 10 km$^2$ and a typical alpine climate
with an average yearly temperature of 1 ºC and yearly precipitation of 2400 mm (Kobierska et al., 2013). Damma

glacier covers 50 % of the watershed and due to climate change has retreated at an average rate of 10 m per year in the last 90 years. However, during 1920–1928 and 1970–1992 the recession was interrupted and the glacier grew, resulting in two moraines (Kobierska et al., 2011). After the retreat of the glacier a soil chronosequence is developed, which has a total length of 1 km (Bernasconi et al., 2008; Bernasconi et al., 2011; Kobierska et al., 2013). The bedrock is coarse-grained granite of the Aare massif and is composed of quartz, plagioclase, potassium feldspar, biotite and muscovite (Schaltegger, 1990). Our study site was extensively described in Bernasconi et al. (2011).

The Göscheneralpsee (Fig. 1b) is a hydropower reservoir of a volume of 75 million $m^3$. A 100 $km^2$ and 20 % glacier covered catchment drains into the reservoir. It includes the watersheds of the Damma, Chelen and Tiefen glaciers and the Voralptal watershed. The Tiefen glacier and Voralptal watersheds do not drain directly into the reservoir but their runoff is redirected through two tunnels. The site is described extensively in Kobierska et al. (2013).

## 3 Model and Data

### 3.1 SWAT model

In this study, we used SWAT 2012 coupled with the ArcView SWAT interface, a GIS-based graphical user interface (Di Luzio et al., 2002) that enables the delineation of the watershed, definition of subbasins, and initial parameterisation. It is a semi distributed, time continuous watershed simulator operating on a daily time step.

Each watershed is divided into subbasins, for which slope, river features, and weather data are considered. Furthermore, the watershed is divided into hydrologic response units (HRUs), which are small surface units with distinctive soil-land use combinations and necessary to capture spatially explicit processes. Each process is simulated for each HRU and then summed up for the subbasin by a weighted average. Subsequently the amount of water, sediment and nutrients that come out from each subbasin enter the respective river.

A modified SCS curve number method is used to calculate the surface runoff for each HRU, based on land use, soil parameters, and weather conditions. The water is stored in four storage volumes: snow, soil moisture, shallow aquifer and deep aquifer. The processes considered within the soil profile are infiltration, evaporation, plant uptake, lateral flow, and percolation. What is important in our study is that melted snow is handled by the model the same way as the water that comes from precipitation regarding the calculation of runoff and percolation. The factors controlling snow melt are the air and snowpack temperature, the melting rate and the area covered by snow. The updated snow cover model takes into account shading, drifting, topography and landcover to create a nonuniform snow cover (Neitsch et al., 2011). Furthermore, runoff from frozen soil can also be calculated by defining if the temperature in the first soil layer is less than 0ºC. Even though the model still allows significant infiltration when the frozen soils are dry, the runoff of frozen soils is larger than that of other soils. A detailed description of the theory behind the model is found in detail in Arnold et al. (1998) and Srinivasan et al. (1998).

Snow processes in high alpine areas are strongly influenced by the terrain features (Ahl et al., 2008; Zhang et al., 2008). Fontaine et al. (2002) revealed the importance of improving SWAT algorithms to include in the model the

influence of elevation and season on the dynamics of the snowpack.. They found that the definition of elevation bands within the model subbasins can significantly improve the performance of the model in watersheds at high altitudes and with large elevation gradients. With the improved snow melting algorithm (Fontaine et al., 2002), streamflow in alpine regions can be successfully simulated by SWAT (Rahman et al., 2013; Grusson et al., 2015; Omani et al., 2017).

**3.2 Input data**

The input data required by SWAT are: topography, soil, land use and meteorological data.

**3.2.1 Topography**

The topography of both study areas was defined using a high precision Digital elevation model (DEM) with 2 m grid cells (swissALTI3D), produced by the Swiss Federal office for Topography (http://www.swisstopo.admin.ch/internet/swisstopo/en/home/products/height/swissALTI3D.html).

**3.2.2 Soil and land use map**

In order to better describe the glacier forefield and to reduce the uncertainty of the calibration for the Damma glacier watershed, detailed soil and land use maps were created based on the observations, field and experimental data from the Biglink and SoilTrEC projects (Bernasconi et al., 2011; Dumig et al., 2011; Andrianaki et al., 2017). The soil map was created by adding new soil types to the SWAT database while the land use classes were based on existing types in the database. For the greater area feeding the Göscheneralpsee, the soil map used, was produced and provided by the Swiss Federal Statistical Office at a scale of 1:200,000 (http://www.bfs.admin.ch/bfs/portal/en/index.html). For land use, we used the Corine land cover dataset 2006 (version 16, 100m resolution) produced by the European Environmental Agency (http://www.eea.europa.eu/data-and-maps/data/corine-land-cover-2006-raster-2).

**3.2.3 Climate data**

Meteorological data from one local weather station and one station of the SwissMetNet network were used. The weather stations are located at the Damma glacier watershed (2025 m a.s.l.) and at Gütsch (2283 m a.s.l.). The meteorological data of the weather Gütsch were provided by MeteoSwiss. The selection of the weather station Gütsch was based on the results of previous research that showed that it has the best correlation in comparison to other weather stations located in the area (Magnusson et al., 2011) with a long enough record for this study. The data from both stations consist of sub-hourly records of air temperature, precipitation, wind speed, relative humidity, incoming short-wave radiation and incoming long-wave radiation from 2007–2013 for Damma weather station and 1981–2010 for Gütsch. The lapse rates for temperature and precipitation, which are very important parameters in SWAT model since they affect snow and glacier melt, and the interpolation methods were based on the findings of Magnusson et al. (2011) who carried out non prognostic hydrological simulations for the Damma glacier watershed. The precipitation and temperature lapse rate parameters of the model are PLAPS and TLAPS and were set to 5 mm km$^{-1}$ and -5.84 $^{\circ}$C km$^{-1}$ respectively.

**Climate change scenarios**: The climate change predictions were provided by the EU regional climate modelling initiative ENSEMBLES (van der Linden and Mitchell, 2009) and were based on the emission scenario A1B. The model chains produced by the ENSEMBLES project are a combination of a general circulation model (GCM) with

a regional climate model (RCM). In Switzerland, model chain data were interpolated to the locations of the
MeteoSwiss stations and the Swiss Climate Change Scenarios CH2011 were created (CH2011, 2011). The delta-
change method was used for the creation of the datasets (Bosshard et al., 2011). Temperature and precipitation
predictions are calculated using daily temperature changes $\Delta T$, and precipitation scaling factors $\Delta P$. Incoming
short-wave irradiation, wind speed and relative humidity were left unchanged. In Switzerland it is predicted that
the mean temperature will increase 2.7–4.1°C and the precipitation during the summer months will decrease 18%–
24% by the end of the century, in the case when no actions for the mitigation of climate change are taken (CH2011,
160 2011).


In this study, three climate scenarios with interpolated data for Gütsch weather station are used. These scenarios
are: the CNRM ARPEGE ALADIN scenario, the ETHZ HadCM3Q0 CLM scenario, which predicts the highest
$\Delta T$ and $\Delta P$ in comparison to the other two, and the SHMI BCM RCA scenario, which predicts the lowest $\Delta T$ and
$\Delta P$, referred to as CNRM, ETHZ and SHMI scenarios respectively. The CNRM, ETHZ and SHMI scenarios were
chosen to be in agreement with the previous study of Kobierska et al. (2013), to be able to carry out a direct
comparison of the three models. The following periods were selected:
Reference period (T0): 1981–2010
Near future period (T1): 2021–2050
Far future period (T2): 2070–2099

Similarly to the predictions for Switzerland, the scenarios for Gütsch weather station predict warmer and dryer
summers and slightly increased precipitation in autumn. The highest $\Delta T$ for the near future period is 1.5°C in the
mid-summer, 2.5°C in late spring, and below 1.0°C in early summer for the CNRM, ETHZ and SHMI respectively
and for the far future period is approximately 5°C in the mid-summer, 4°C along the whole summer and 3°C in
early summer respectively. The biggest temperature increase is predicted at the end of the century when the
strongest agreement between the different model chains is observed. Precipitation changes for the near future
period are within the natural variability apart from a clear trend in dryer summers. The trend of dryer summers is
most prominent for the far future period. Furthermore, most model chains predict slightly higher precipitation in
autumn. The average $\Delta P$ value for the near future period is 1.0 and for the far future period is 0.99. The climate
change data were also used for different sites in the Alps (Bavay et al., 2013; Farinotti et al., 2012).
**3.2.4 Runoff data**
Runoff of the Dammareuss stream that drains the Damma glacier watershed was measured every half an hour at a
gauging station at the outlet of the watershed (Magnusson et al., 2011). The runoff of the total area that feeds the
Göscheneralpsee is the inflow of the reservoir and the data from 1997–2010 were provided by the energy company
responsible for the management of the reservoir.
**3.2.5 Glacier extent**
Data on the glacier extent for the present period but also for the two periods of the climate change scenarios were
provided by Paul et al (2007). They estimated the evolution of the Swiss glaciers by using hypsographic modelling,
based on the shift of the equilibrium line altitude. However, SWAT is not a model that considers glacier flow
dynamics and therefore, in this study, the glaciers were incorporated in SWAT as the initial snow content in each
subbasin and for each elevation band. The initial snow is given as the snow water equivalent in mm instead of
snow as the density of snow can be variable. For this reason, the calculation of the snow water equivalent was
conducted by considering an average density of ice.
**4 Methodology**
The purpose of this study is to assess the transferability of SWAT in temporal and spatial scales at a high altitude
alpine and glacierised site. This way it is tested whether the model can be transferred and is capable for the
simulation of runoff but also for further climate change studies on an ungauged glacierised watershed.
Furthermore, this methodology tests its robustness under these extreme climatic and geographical conditions. For
this reason, SWAT was initially calibrated for the small Damma watershed, which is well monitored through the
CZO projects, and then it was upscaled and applied for the greater area feeding the Göscheneralpsee reservoir and
includes the Damma glacier watershed. The upscaling of the model was verified by comparing model results with
the reservoir data provided by the managing company.

Since the Damma glacier watershed is part of the greater Göscheneralpsee feeding catchment, the parameters of
the model were transferred using the spatial proximity approach, with no further regionalisation procedure. In this
case, the initial setup of SWAT for the greater catchment was conducted using the input data presented in section
3.2 and only the parameters presented in Table 1 were changed to the calibrated values derived from the calibration
of the Damma glacier watershed. The initial parameterisation of the model during the setup and the watershed
delineation assisted in the transferability of the model since a number of parameters is already defined based on
the topography, land use and soil data.

Subsequently, in order to assess its transferability on a temporal scale, climate change simulations were conducted
and results were compared with those of a previous study for the same area, which used two other hydrological
models with different characteristics, PREVAH and Alpine 3D (Kobierska et al., 2013).

This methodology is a modified version of the proxy-basin test introduced by Klemeš (1986), which is one of the
proposed testing schemes for the enhancement of the calibration and validation procedure in hydrological
modelling. According to Klemeš (1986) the proxy basin test can be used to test the geographical transposability
of the model between two regions, for subsequent simulation of the streamflow in ungauged watersheds with
similar characteristics. The model is calibrated and validated for two different but similar watersheds and if the
results are acceptable it is then considered safe to be transferred and used at a third watershed with similar
characteristics.
**5 Model setup, calibration and validation**
SWAT was initially setup for the Damma glacier CZO and the greater area feeding the Göscheneralpsee using the
topography, soil and land use data presented in section 3.2. Following the delineation procedure, the Damma
watershed and the greater area were divided into 5 and 25 subbasins respectively. By setting the lowest possible
thresholds for land use, slope and soil, 48 HRUs were created for Damma watershed and 285 HRUs for the greater
area. Finally, six elevation bands were defined for each subbasin of both study sites. The setup was complete with

the addition of the meteorological input and the definition of the initial snow for each elevation band of each subbasin. For the climate change simulations, the meteorological input consists of the climate change scenarios described in section 3.2.3 and the initial snow that corresponds to the first year of each future period, as calculated by the glacier extent data described in section 3.2.5.

**5.1 Model calibration**

SWAT was calibrated for the Damma watershed only, using the meteorological data from 2009 to 2011 and validated with the data from 2012 to 2013. Data for the years 2007 and 2008 were used for the warm-up and the stability of the model. For the better identification of the parameters that influence the hydrology of the site the calibration was first conducted manually. The most sensitive parameters during this step were related to snow melt such as: i) TIMP, the snow pack temperature lag factor, ii) SMFMX, the snow melt factor on the 21st of June (mmH$_2$O / ℃ day$^{-1}$), iii) SMFMN, the snow melt factor on the 21st of December (mmH$_2$O / ℃ day$^{-1}$), CN_FROZ, which was set to active in order and finally the snow fall and snow melt temperatures SFTMP and SMTMP respectively. Because most of the subbasins of the Damma glacier watershed, delineated during the initial setup of the model, were partially glacier covered, it was decided to follow a simple approach and apply the same snow parameters for all the subbasins. This means that the same parameters were applied for both glacier and snow dynamics.

Groundwater flow parameters such as the GW_DELAY, the groundwater delay time, ALPHA_BF, the base flow alpha factor and the SURLAG, the surface runoff lag coefficient, were also found to play an important role on the performance of the model. Evapotranspiration (ET) related parameters were not significant since our study site is above the tree line and ET is relatively minor.

The manual calibration was followed by an automatic calibration and uncertainty analysis using the SWAT-CUP software with the Sequential Uncertainty Fitting ver. 2 (SUFI-2) algorithm for inverse modelling (Abbaspour et al., 2007). Starting with some initial parameter values, SUFI-2 is iterated until (i) the 95% prediction uncertainty (95PPU) between the 2.5th and 97.5th percentiles include more than 90% of the measured data and (ii) the average distance between the 2.5th and 97.5th percentiles is smaller than the standard deviation of the measured data. A model is considered calibrated when the chosen criterion between the best simulation and calibration data reaches the best value (Abbaspour et al., 2007). The parameters introduced in SWAT-CUP as well as their range are the ones that were identified during the manual calibration as the most important.

The criterion used for the calibration with SWAT-CUP is the Nash-Sutcliffe (Nash and Sutcliffe, 1970) model efficiency (NS), since it was the criterion available in SUFI-2 that is commonly used in hydrological studies. The NS shows the relationship between the measured and the simulated runoff (Eq. 1). The performance of the calibrated model was further evaluated by the square of Pearson's product moment correlation R$^2$, which represents the proportion of total variance of measured data that can be explained by simulated data. Better model performance is considered when both criteria are close to 1. NS coefficients greater than 0.75 are considered ''good,'' whereas values between 0.75 and 0.36 as ''satisfactory'' (Wang and Melesse, 2006).

$$NS = 1 - \frac{\Sigma(y - \hat{y})^2}{\Sigma(y - \bar{y})^2}, \tag{1}$$

where $y$ is the individual observed value, $\hat{y}$ for the individual simulated value and $\bar{y}$ the mean observed value.
However, as Schaefli and Gupta (2007) pointed out, the NS criterion is not enough to judge the efficiency of the
model when simulating runoff with high seasonality like the one in high altitude watersheds. Therefore, as an
additional criterion for the performance of the model, a benchmark efficiency indicator was calculated, according
to Eq. 2:
$$BE = 1 - \frac{\sum_{t=1}^{n}[q_{obs}(t) - q_{sim}(t)]^2}{\sum_{t=1}^{n}[q_{obs}(t) - q_b(t)]^2}, \tag{2}$$

where $q_{obs}$ is the observed runoff; $q_{sim}$ is the simulated runoff by SWAT; and $q_b$ is runoff given by the benchmark
model. The calendar day model was chosen as benchmark (Schaefli and Gupta, 2007), which is the observed
interannual mean runoff for every calendar day.

Table 1 shows the default and the after calibration values of the SWAT parameters that were changed during
calibration. TIMP was set to a very low value indicating that the glacier is not affected by the temperature of the
previous day as much as the snowpack would be. Snow and glacier melt in Damma watershed occurs from April
to September, a fact that explains the low value of the SMFMN parameter (0.1 mmH$_2$O / °C-day), the minimum
melt factor, while the SMFMX is set to the value of 4.7 mmH$_2$O / °C-day. SURLAG and GW_DELAY play an
important role in the model performance as they control the melted snow routing process and the hydrologic
response of the watershed. Damma glacier watershed has a fast response and therefore GW_DELAY was set to
0.5 days. SMTMP is also sensitive since it is the controlling factor for the initialisation of the snow melt,
considering the availability of snow for melting on a specific day. As a result, model-generated peak runoff is
significantly influenced by the variation in SMTMP. Finally, ALPHA_BF was set to value 0.95, which is a typical
value for a fast response watershed.

The results of the calibrated model for the daily runoff and the observed data are presented in Fig. 2(a), while
cumulative runoff is presented in Fig. 2(c). The fit of the model to the observed data is satisfactory and the results
of the calibrated model matched the observed data throughout most of the year. The graph of the cumulative runoff
(Fig. 2c) shows that runoff is slightly overestimated in July and August, when it is dominated by glacier melt. Best
results occur for the years 2009 and 2010. 2011 is characterised by unusually warm and dry months of September,
October and November which resulted in a slight underestimation of the runoff. Overall SWAT performance for
the calibrated period is considered very satisfactory since the NS efficiency is 0.84 and $R^2$ is 0.85. BE for this
period is 0.22, a value that we consider to be satisfactory and is comparable to that of the previous model, calibrated
for the greater area of Göscheneralpsee.

**5.2 Sensitivity analysis**

The automatic global sensitivity analysis was conducted with SWAT-CUP software and 17 input parameters were
analysed. It revealed that the most sensitive parameters are the same as the ones observed during manual
calibration. More specific the most sensitive ones in descending order are TIMP, GW_DELAY, SMTMP,
SMFMX, ALPHA_BF and SURLAG with p values 0 for TIMP and very close to 0 for the remaining parameters.
The least sensitive parameters were left to their default value.

**5.3 Model validation**

SWAT was validated using the meteorological data for 2012 and 2013 and the results of the model as well as the measured runoff are presented in Fig. 2**(b).** Figure 2(d) presents the cumulative graphs. The SWAT model for this period performed efficiently, similarly to the calibration period, with a Nash-Sutcliffe efficiency of 0.85, $R^2$ 0.86.and the BE 0.25. A small inconsistency is observed in the late spring of 2012, when estimated runoff is underestimated, probably due to the extremely wet May in that year that cannot be efficiently simulated. Although, due to the lack of longer monitoring data, the total calibration-validation period 2009-2013 is short, it still includes a relatively large variability in the weather conditions and precipitation amounts and despite this variability the overall model performance is very satisfactory. The small seasonal differences in model performance are due to the evolution of runoff generation throughout the season: runoff in spring and early summer (May, June) comes mainly from snowmelt while in July and August it stems mainly from glacier melt. Although there are two different water sources during the two different periods, we can only assign one set of parameters. We can nevertheless conclude that SWAT can be successfully applied for a partly glacierised watershed.

**6 Results and Discussion**

**6.1 Upscaling SWAT to the greater catchment feeding the Göscheneralpsee reservoir**

The results of the model for the greater area that feeds the Göscheneralpsee, are presented in Fig. 3(a) together with the measured inflow in the reservoir. The observed and predictive cumulative flow is presented in Fig. 3(b). Model performance criteria were lower than for the calibration period as NS dropped to 0.49 and the $R^2$ to 0.72. The cumulative graph shows that there is an overall good agreement between model results and the measured reservoir inflow. Both Figures 3(a) and 3(b) show that there is an overestimation of total runoff for the period 1999-2002, which appears to be linked to the higher precipitation amounts during this period. Measured precipitation measured at Gütsch weather station for this period is up to 46 % higher than the average precipitation of 1981-2010.

The predictability of the model was further tested by analysing key parameters related to median runoff such as spring snowmelt timing, timing of peak flow, autumn recession period and the centre of mass (COM), which can indicate temporal shifts in the hydrological regime. Table 2 shows the difference in days between the observed and simulated values of the above parameters for each year of the period 1997-2010. A 15day moving average window was applied to daily runoff. Snowmelt timing and autumn recession are predicted successfully since the differences for most years are zero or close to zero, except for 2000 and 2002 for autumn recession. Peak flow timing shows some inconsistencies between observed and simulated data for certain years, which are mainly related to the fact that for these years and during the snowmelt period, SWAT produces results with higher peaks. Finally, the COM of the simulated data is in good agreement with that of the observed data, with an average difference of 4 days.

On the whole, SWAT performance is considered to be satisfactory and it was successfully transferred to the greater Göscheneralpsee feeding catchment. One of the main reasons for the deterioration of the model performance during the years with higher precipitation, 1999-2002, is that SWAT doesn't differentiate between snow and glacier dynamics and only one parameter for both snowmelt and glacial melt rate was applied. This becomes more

important in our study, since there is a difference between the percentage of glacial coverage of the two catchments,
with the Damma glacier watershed being 50% covered while the greater catchment 20%. In Omani et al. (2017)
this issue was partly addressed by applying different snow parameters to the glacier covered subbasins. However,
the subbasins in our calibration watershed, the Damma glacier watershed, were partly glacierised and for this
reason it was decided to apply only one set of snow parameters for the whole watershed.

Furthermore, some inconsistency is caused by the fact that for the two out of the four of the watersheds of the
greater area feeding the Göscheneralpsee, runoff is drained through tunnels into the reservoir. In addition, there is
a difference in the hydrological response between the Damma glacier watershed in comparison to the greater area.
Damma is characterised by very steep slopes (even up to nearly 80 degrees) and runoff originates mainly from
snowmelt, glacial melt and rainfall (Magnuson et al., 2012). The ALPHA_BF parameter of SWAT was set to a
high value and the GW_DELAY to low, parameter trends that characterise a watershed with a high response. On
the other hand, the Göscheneralpsee feeding area is less steep on average. The combination of these two factors
might be the reason, why some of the simulated peaks are higher but also narrower compared to the observed
inflows into the reservoir.

Finally, SWAT results were compared to results from PREVAH and ALPINE3D models, already published in
Magnusson et al. (2011) and Kobierska et al. (2013) (Fig. 4). PREVAH is a semi-distributed conceptual
hydrological model suited for applications in mountainous regions (Viviroli et al., 2009a; Viviroli et al., 2009b)
while ALPINE3D is a fully-distributed energy-balance model (Lehning et al., 2006).

Figure 4 shows the interannual average of the period 1997-2010 daily runoff for each model. SWAT overestimated
the runoff of the snowmelt period, from May to the beginning of July, while from mid July to late September its
results are close to the observed values and in agreement with the other two models. Finally, in October runoff is
slightly underestimated. The seasonality in variation between model results and observed values is linked to the
application of only one melt rate for both snowmelt and glacial melt periods. The best fit of the model is observed
when glacial melt is the major contributor to runoff, while it is overestimated during the snowmelt period, which
is the reason of the excessive simulated runoff during the 1999-2002 period of high precipitation (Fig.3), as
discussed above. Seasonal variability in model performance is observed not only for SWAT but also for
ALPINE3D and PREVAH, as ALPINE3D underestimated runoff during the snowmelt period, from May to June,
while on the other hand runoff was slightly overestimated by PREVAH in October and November (Kobierska et
al., 2013).

Furthermore, a combination of the factors discussed above about the applied snowmelt parameters and the
deviation in hydrological response between the two areas because of human intervention and
topographical/geographical features is the reason why SWAT doesn't simulate efficiently the winter low flows.
**6.2 SWAT transferability on a temporal scale**
As a next step, we assessed whether SWAT can be transferred at a temporal scale, by running climate change
scenarios for the greater area that feeds the Göscheneralpsee. In order to verify the model transferability, results
were compared with the climate change study in Kobierska et al. (2013) using the same time periods as follows:
Reference period (T0): 1981–2010
Near future period (T1): 2021–2050
Far future period (T2): 2070–2099

The results of SWAT model are presented as the interannual average runoff for each different scenario along the
whole period in Fig. 5(a) for the near and in Fig. 5(b) for the far future periods.

During the reference period runoff peaks in early July when snowmelt is combined with glacier melt. For the near
future period T1, the main difference happens from July to September when runoff is dominated by glacier melt.
During this period, predicted runoff for all scenarios, and in particular for the warmer ETHZ scenario, is lower
than the reference period, indicating that the glacier melt cannot compensate the predicted decrease in precipitation.
From September until the end of the season, simulated stream flow of all scenarios is higher than the reference
period, which is explained by the higher predicted precipitation during autumn. The annual peak remains in early
July, since the glacier has not melted away yet, providing glacier melt.

For the far future period T2, runoff from spring to June is predicted to increase significantly for all three scenarios
due to more intense snowmelt. In addition, higher precipitation is predicted by the climatic data for this period.
Based on the available glacier extent data described in section 3.2.5, we estimated that in 2070, the total glacier
volume will be reduced to almost half, resulting in less glacial melt between July and late August. For this reason,
and in combination with the significant decrease in precipitation, predicted by all scenarios for this period, the
simulated runoff is lower than that of the reference. Finally, the snow free period will extend until December
instead of September.

At the end of the T2 period, the average temperature increase in our site will be 3.35°C and only a small part of
the glacier will remain in high elevation. The date of peak flow will shift to be in the beginning of June. The main
runoff volume is expected to be observed in spring and early summer while during the glacier melt period,
streamflow is significantly lower than that of the reference period. Overall the total water yield for the scenarios
in T2 period is predicted to decrease.

To better observe the seasonal changes of estimated runoff, Fig. 6 shows the interannual average runoff for a)
May-June, b) July-August and c) September-October for the T1 and T2 future periods divided by the average of
the reference period of the same months for all the three scenarios. In May and June, as mentioned above, runoff
is mainly dominated by snowmelt. The three climate change scenarios predict increased temperatures and higher
precipitation during May and June which result in faster snowmelt and therefore in the increased predicted runoff,
as observed in Fig. 6(a). The increase is higher in the far future due to the higher temperatures. The only exemption
to that is the SHMI scenario for the near future period, since it is the colder scenario that predicts the lowest
temperature and precipitation changes. In July and August, climate change scenarios predict a significant decrease
in precipitation, which is also depicted in the predicted runoff. The scenario that has the most drastic effect is the
ETHZ because it is the scenario that predicts the highest increase in the temperature and decrease in the
precipitation. For September and October, results do not show a clear trend for the warmer ETHZ scenario,
however for the CNRM and SHMI scenarios, predicted runoff is lower than the reference. Finally the predicted
runoff of the far future period T2 shows higher fluctuations from year to year than that of the near future period
especially from September to October.

The climate change predictions of SWAT and the subsequent conclusions show many similarities in the seasonal
variations with that of ALPINE3D and PREVAH. This observation is very promising since it demonstrates that
SWAT could be applied to climate change studies in ungauged high altitude watersheds. There are however
uncertainties and differences between the models. ALPINE3D and PREVAH models predict the spring peak flow
to shift approximately by 3 and 6 weeks for the near and far future periods respectively. On the other hand, the
shift in peak flow with SWAT is smaller and especially for the near future period a 10 day shift is predicted only
with the warmer ETHZ scenario (Fig. 5).
**7 Conclusions**
This study is an assessment of the transferability or upscaling of SWAT on a spatial and temporal scale for a partly
glacierised catchment at a high altitude. For this reason, we followed an approach similar to the proxy-basin test
introduced by Klemeš (1986).

Firstly, SWAT was calibrated and validated for the Damma glacier watershed and it was demonstrated that despite
the extreme conditions of this high alpine watershed, SWAT performed successfully, with satisfactory NS and BE
efficiencies. Subsequently, we assessed the transferability of the model by upscaling and applying SWAT for the
greater area that drains into the Göscheneralpsee reservoir and includes the Damma glacier watershed. By
comparing model results with existing inflow data, we showed that the model was able to predict key parameters
such as the snowmelt timing, autumn recession period and the peak flow timing. However, overestimation of
runoff during the snowmelt period, especially in wet years, highlights the importance of taking into account the
difference in snow and glacier dynamics. It showed that better performance could have been achieved if different
parameters for snow and glacial melt had been applied. This observation is quite important for study sites where
streamflow is greatly dependent on both snow- and glacier melt. Model performance was potentially affected in
the greater catchment due to hydropower infrastructure such as tunnels.

The temporal transferability of SWAT was analysed by assessing the impact of climate change on the hydrology
of the greater catchment and comparing these results with a previous climate change study conducted for the same
area. Climate change predictions showed that the hydrological regime will change significantly in the future
especially towards the end of the century. Daily runoff during May and June is predicted to increase because more
intense snowmelt and the predicted wetter springs. Projected runoff from July to October, mainly for the far future
period but also for the near future, is significantly decreased. These results show many similarities with those
previously published.

In conclusion, our findings indicate that SWAT is a model that can be successfully transferred to simulate
streamflow and climate change impact for high altitude glacierised ungauged watersheds. The upscaling
methodology used here, where SWAT is calibrated for a small watershed and then applied for a greater area that
includes the calibration watershed, is a simple but still effective approach. It can be valuable in predicting
streamflow of ungauged watersheds, in large scale hydrological simulations and for policy makers working in
water management.

**Author Contributions**

Maria Andrianaki applied SWAT model, analysed data and prepared the manuscript with contributions from all
co-authors. Juna Shrestha reviewed the manuscript and assisted in the modelling procedure. Florian Kobierska
provided meteorological and runoff data. Nikolaos P. Nikolaidis provided guidance for the research goals. Stefano
M. Bernasconi was the supervisor of the research project and provided the funding that lead to this publication.

**Competing interests**

The authors declare that they have no conflict of interest.

**Acknowledgements**

This study was supported by the European Commission FP 7 Collaborative Project: Soil Transformations in
European Catchments (SoilTrEC) (Grant Agreement No. 244118). We thank Thomas Bosshard (Institute for
Atmospheric and Climate Science, ETH Zürich, Switzerland), Frank Paul (Department of Geography, University
of Zürich, Switzerland), MeteoSwiss and SwissTopo for providing all the necessary data for the completion of this
study. We would also like to thank the Editor and the reviewers for their valuable contributions in improving this
manuscript.

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

**Table 1 The default and calibrated values of the most sensitive SWAT parameters**

| Parameter | Unit | Cal. Value | Default |
|---|---|---|---|
| SFTMP | ºC | -0.5 | 1 |
| SMTMP | ºC | 2.5 | 0.5 |
| SMFMX | mm $H_2O$ / ºC day$^{-1}$ | 4.7 | 4.5 |
| SMFMN | mm $H_2O$ / ºC day$^{-1}$ | 0.1 | 4.5 |
| TIMP | | 0.011 | 1 |
| SURLAG | | 0.001 | 4 |
| | | | |
| CNCOEF | | 0.5 | 1 |
| SNOCOVMX | mm $H_2O$ | 500 | 1 |
| SNO50COV | % | 0.3 | 0.5 |
| | | | |
| ALPHA_BF | days | 0.95 | 0.048 |
| GW_DELAY | | 0.5 | 31 |
| GW_REVAP | | 0.02 | 0.02 |
| | | | |
| LAT_TTIME | | 0.0001 | 0 |
| CN2 | | 35 | Na |
| SLSOIL | m | 5 | Na |
| ESCO | | 1 | 0.95 |
| | | | |
| SOL_AWC | mm $H_2O$/mm soil | 0.05 | Na |



**Table 2 Absolute difference in days between simulated and observed values of the snowmelt timing, autumn recession**
**period, peak flow timing and the centre of mass (COM), for the greater catchment feeding the Göscheneralpsee.**

| Year | Snowmelt timing | Autumn recession period | Peak flow timing | COM |
|---|---|---|---|---|
| 1997 | 0 | 1 | 48 | 7 |
| 1998 | 2 | 1 | 2 | 4 |
| 1999 | 4 | 0 | 27 | 1 |
| 2000 | 0 | 16 | 19 | 3 |
| 2001 | 0 | 1 | 1 | 1 |
| 2002 | 0 | 19 | 0 | 8 |
| 2003 | 2 | 5 | 2 | 1 |
| 2004 | 1 | 4 | 21 | 2 |
| 2005 | 1 | 0 | 1 | 4 |
| 2006 | 3 | 1 | 3 | 4 |
| 2007 | 3 | 1 | 7 | 8 |
| 2008 | 2 | 0 | 2 | 3 |
| 2009 | 1 | 0 | 13 | 5 |

| | | | | |
|---|---|---|---|---|
| 2010 | 2 | 0 | 1 | 6 |
| Average | 1.4 | 3.5 | 11.0 | 4.0 |

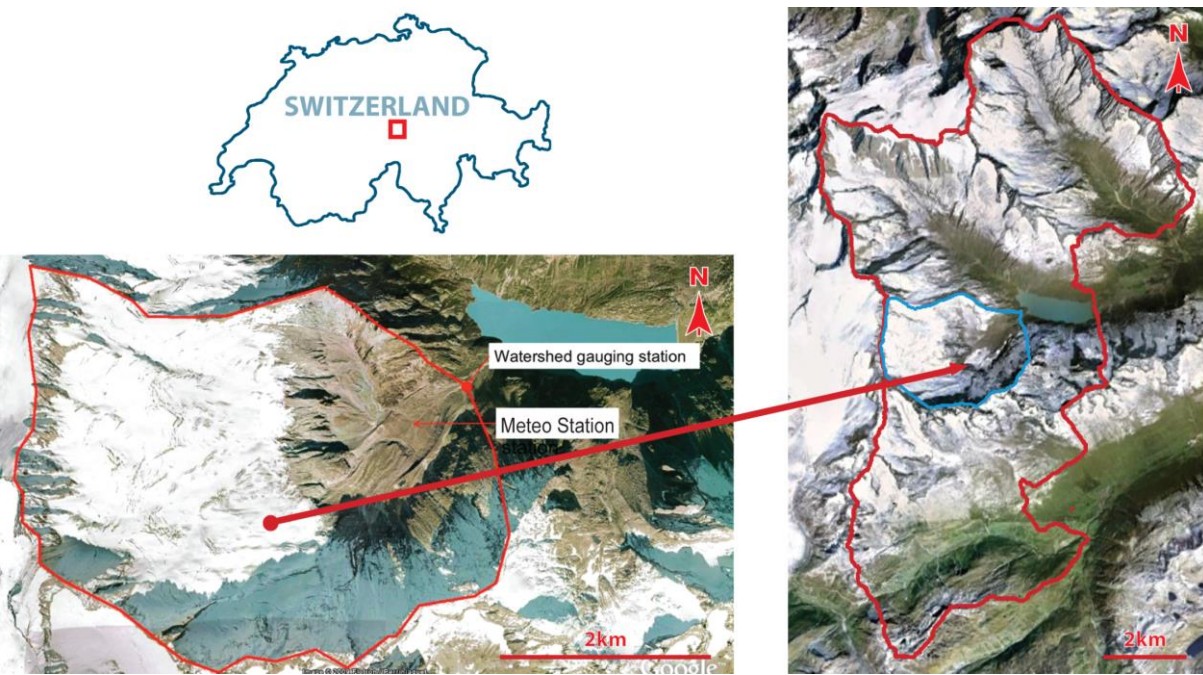

**Figure 1 Map showing the Damma glacier watershed on the left and the greater area that feeds the Göscheneralpsee on the right.**


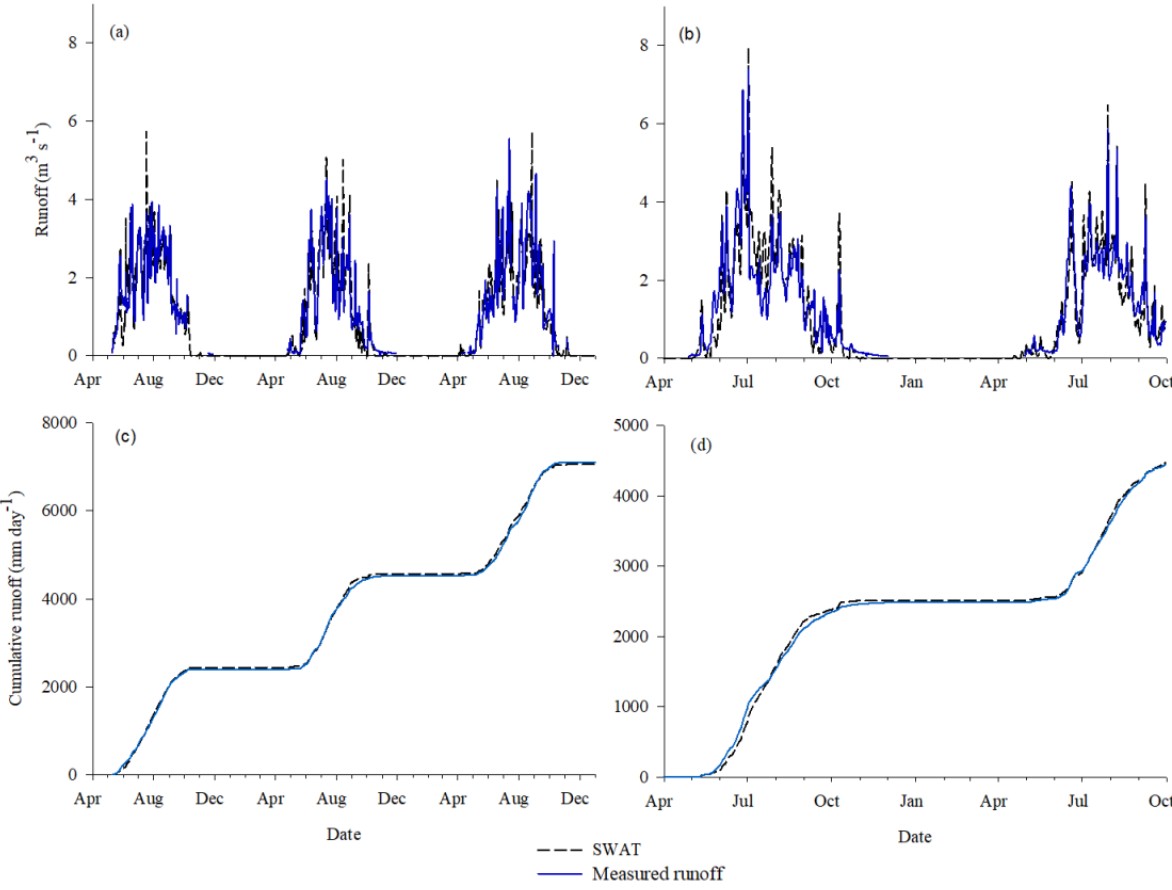


**Figure 2 Observed and cumulative runoff for a) and c) the calibration period 2009-2011 and for b) and d) the validation**
**period 2012-2013**


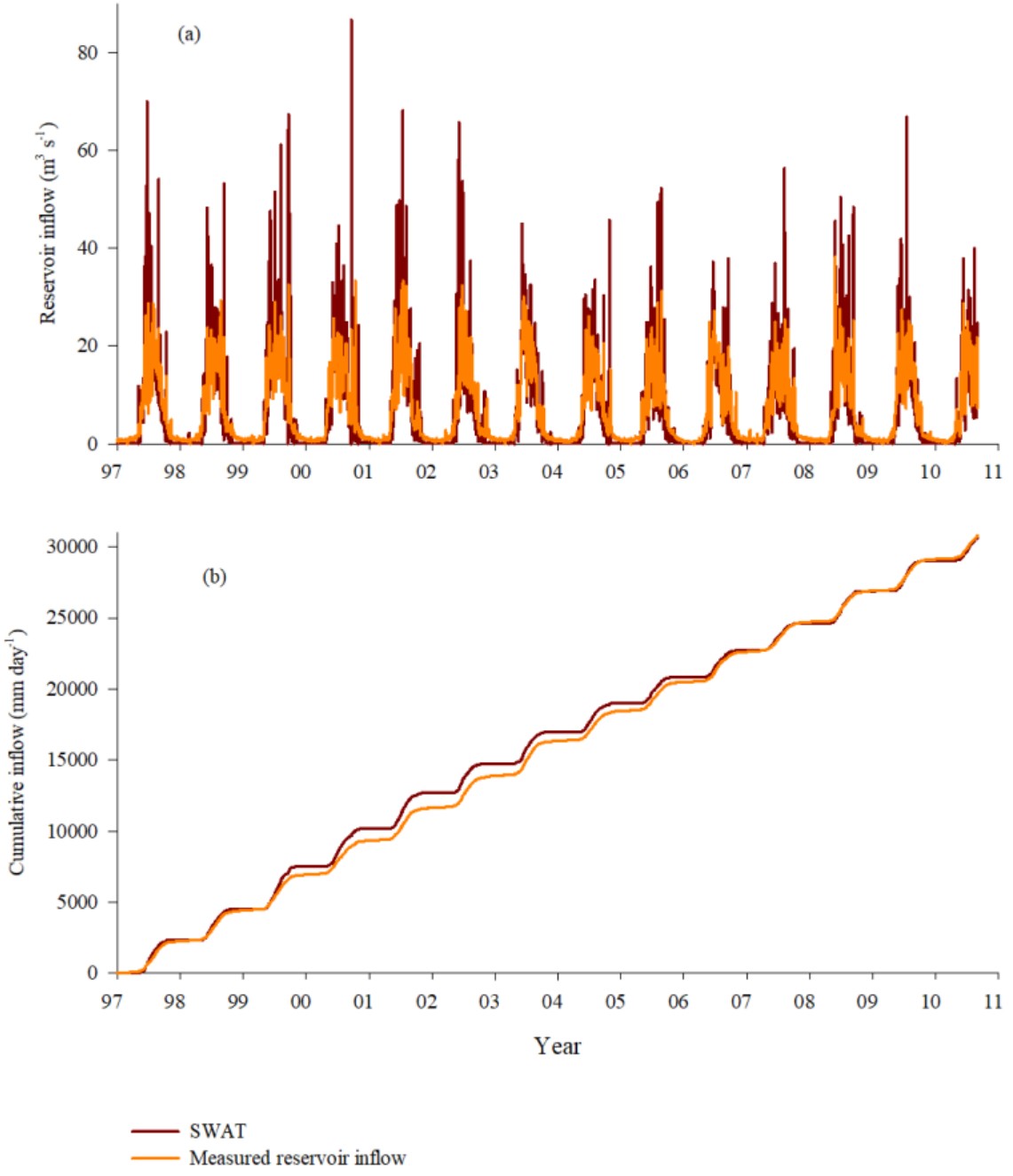


**Figure 3 SWAT results and measured inflow of the feeding catchment of the Göscheneralpsee reservoir for the period**
**1997-2010. Graphs in (b) show the observed and simulated cumulative runoff over this period.**



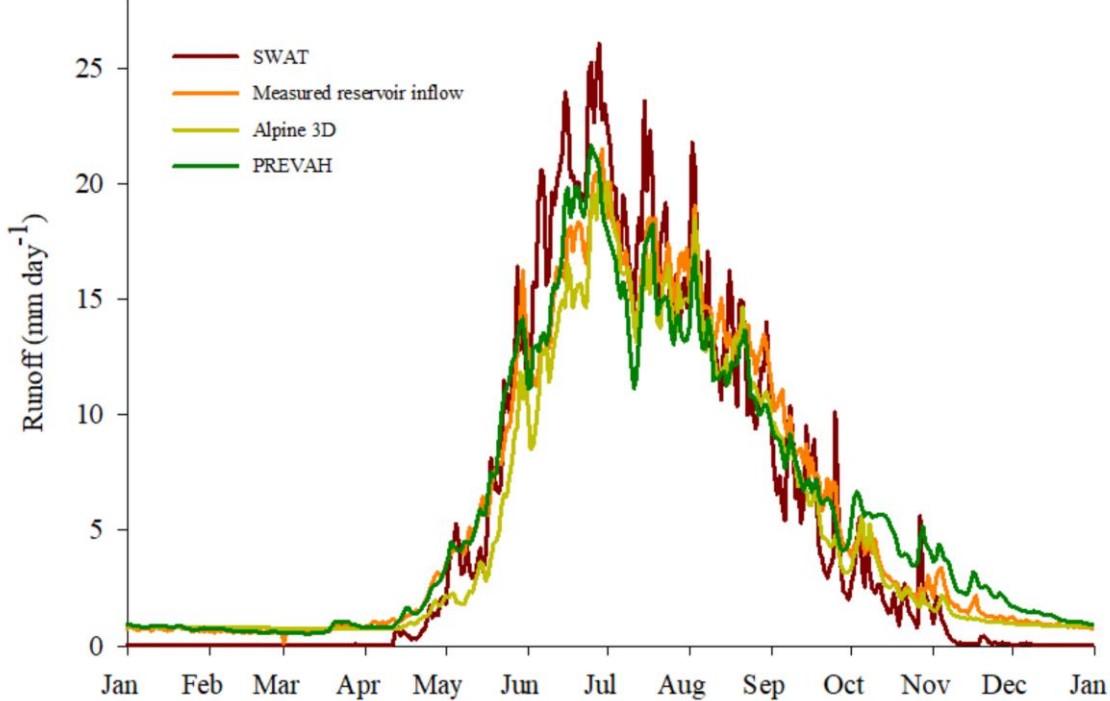


**Figure 4 Interannual average of the results of SWAT, ALPINE3D and PREVAH models and the measured runoff of the Göscheneralpsee feeding catchment for the 1997-2010 period.**



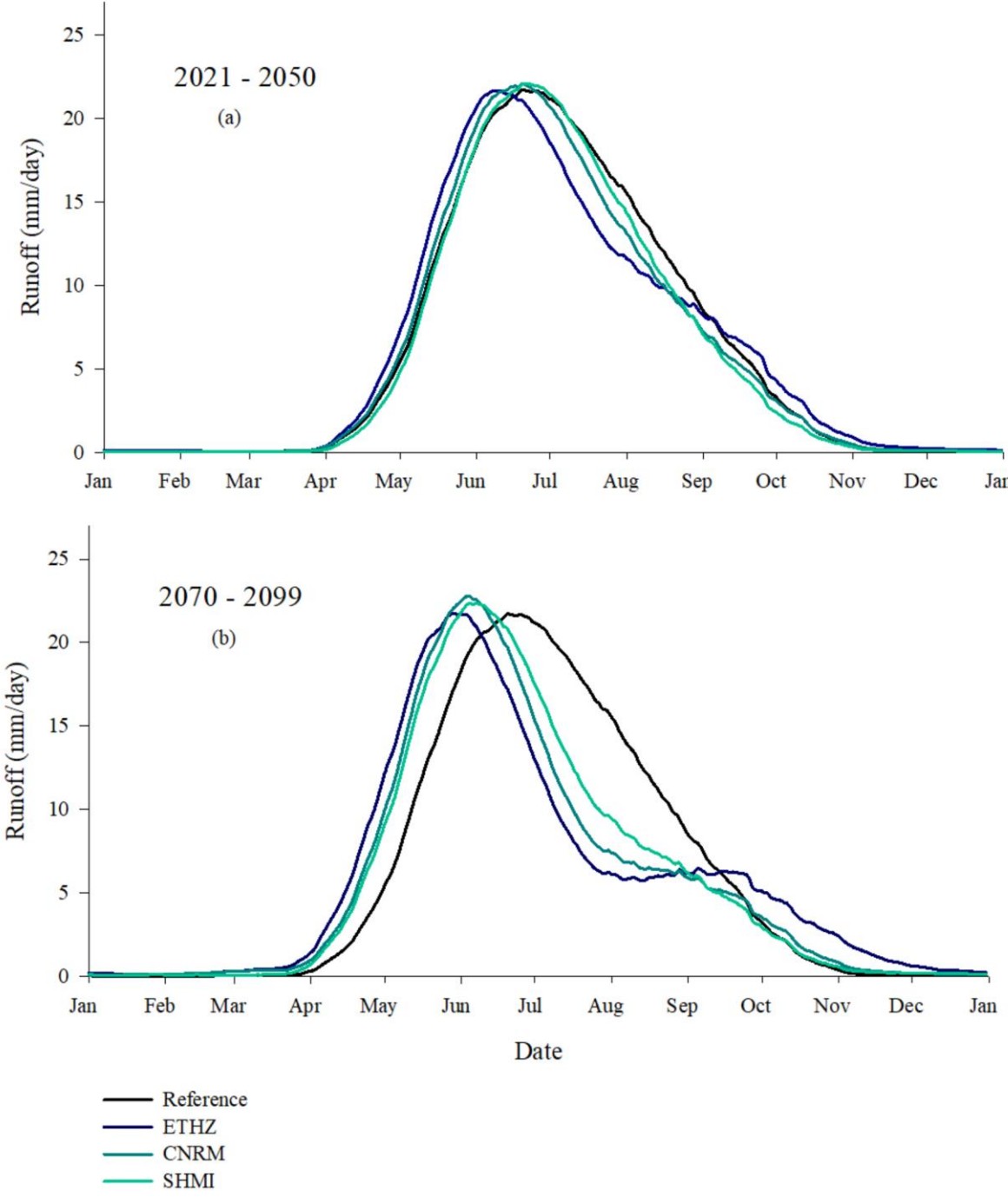


**Figure 5 Interannual average of SWAT results of the three climate change scenarios and the reference period T0 for**
**the Göscheneralpsee feeding catchment a) for the T1 period 2021-20150 and b) for the T2 period 2070-2099. A 30 day**
**average window is applied.**

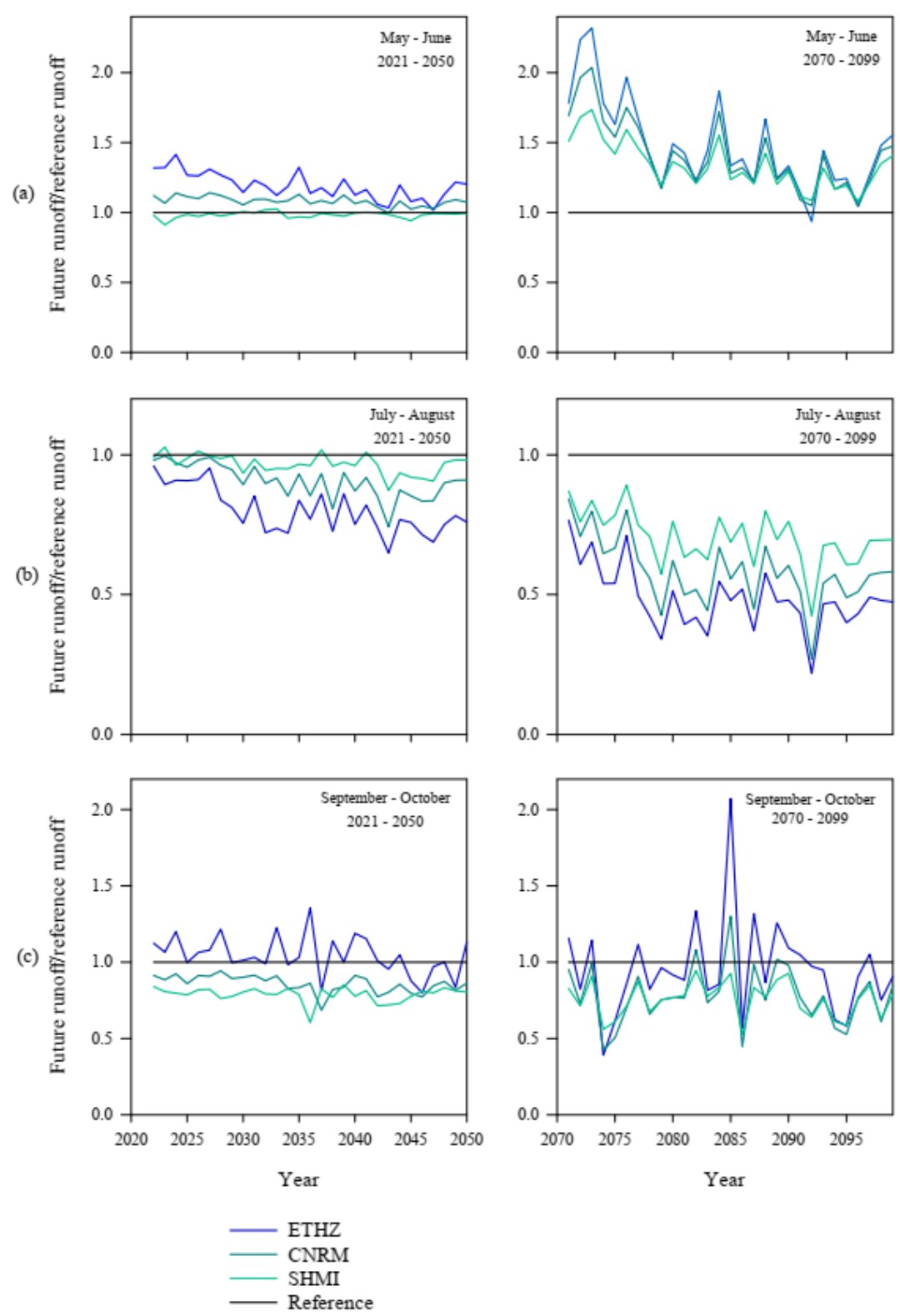


**Figure 6 Seasonal changes of the simulated with SWAT runoff of the Göscheneralpsee feeding catchment for the reference T0 and future periods T1 and T2 for all three climate change scenarios. The interannual mean of the months a) May and June, b) July and August and c) September and October is taken.**
