# Peer review of "Assessment of SWAT spatial and temporal transferability for a high altitude glacierised catchment"

_Hydrology and Earth System Sciences, 2018_

## Referee Comment (RC1) · G. Thirel (Referee) · 16 Nov 2018

The paper by Andrianaki et al. deals with a topic of interest for HESS readers: the modelling of runoff in a glacierised catchments and projections of its evolution. The manuscript reads easily and is concise; I would like to thank the authors for that, as it is often not the case and readers are burdened with loads of not so useful information in many papers.

That said, I feel that there is room for improvement before the paper reads as a scientific paper. Here are my **main remarks**.

1) The main criticism is that I failed to identify clearly what the readers could bring home from this manuscript. Definitely not a new methodology, as the SWAT model is basically used as is, the sensitivity test is not detailed and the calibration and climate change exercises are classical. In my opinion, results are also not so remarkable. It is very interesting to see the validation exercise on a different period and then on a different catchment, but in the end we have results about one catchment and the calibration period is very short. As a consequence, we could wonder if we have the right answer for the right reason or not. I find it very difficult to extrapolate anything from results on this catchment for further works.

   If the main additional value of the paper is the fact that SWAT works for this area, then this should be better highlighted and put into perspective with relevant literature. This reflects on the objectives of the study, which are barely presented in the paper and makes it look like an application of the model rather than an actual research work. Only lines 51-52 give some elements on the interest of this work. Consistently, the conclusions only briefly highlight one novelty of the study (L. 354).

   In my opinion, the abstract, the introduction and the conclusions should be clear about the novelty of this work.

2) It is, if I'm not wrong, never clearly stated that calibration of SWAT is done compared to discharge observations only. Calibration is mentioned many times (abstract, end of introduction, section 3.3) but the used observation is not given. SWAT is physically based and snow observations are definitely an additional value to models calibration in snowy areas, so it is legitimate to wonder if the authors used any kind of snow data here.

3) The calibration set up is unclear and at some point flawed to me.
   First, we don't know exactly what the objective function is: authors introduce NS and $R^2$ but they don't specify how they used them: through a composite criterion? With a Pareto front? Then, the use of NS in snowfed basins is not advised. Indeed, this criterion relates the performance to the mean observed discharge, which is a bad predictor in such a seasonally-variable environment (see Schaefli and Gupta (2007)). It also underestimates discharge variability.
   Finally, we don't know how the parameters from the small basin are transferred to the larger one. Are some of these parameters time or scale dependent? It is just said that they are adjusted.

4) The structure of section 4.1 is not easy to follow. Some kind of sensitivity test is done to identify which parameters to calibrate. I failed to understand if it was done by the authors, and if yes I don't understand why it is mentioned only in the third paragraph, so after talking about the values of the calibrated parameters. Also, the word "set" is often used to refer to

parameters; as it is unclear what is meant since both a manual calibration and an automatic one are mentioned, I got a bit lost.

In addition, authors seem to infer that Table 1 shows the results of a sensitivity test. What I rather see here is how different the calibrated values are from the default ones, some of them being unrealistic maybe (I don't know where they come from). L. 239: which ones are the least sensitive ones?

5) The actual setup of this whole study is not justified by the authors. Why is the model calibrated on the small basin that has few data and validated on the large basin with a lot of data rather than the opposite?

6) L. 304: I thought that the black (reference) curve in Fig. 7 should be the same as the SWAT curve in Fig. 6, but it does not seem so. Did I get something wrong? The resolution of Fig. 7 could be improved, it is more difficult to read than Fig. 6.

7) L. 317: the authors state that the volume of the glacier reduces to half in 2070. I wonder how this is considered in the SWAT model. Indeed, I expect that the initial conditions of the model (due to the Delta method used for producing the climate projections a continuous hydrological projection cannot be done) had to be adjusted. How was that done? Also, please precise who estimated this reduction (authors? Literature?).

**Minor remarks:**

Title: The title is not very sexy… Also CZO is an acronym, is it well known enough to be used in a title?

L. 30, 32 and many other places: a space is missing after the semi-colon.

L. 31: I think that the lack of observed data of sufficient quality could also be mentioned.

Section 2: what is the surface area of the small watershed? It is only given for the larger one.

L. 60: after "(Fig. 1)" I think that "is" is missing.

L. 62: inconsistent (lack of) space between number and unit.

L. 69, 74…: why is "et al." suddenly in italics?

L. 77: I would add a comma after "interface"

L. 135: strange punctuation after "Climate change scenarios"

L. 149-150: are the parentheses necessary around Delta P and Delta T? "(Bosshard et al. 2011)" should be "Bosshard et al. (2011)"

L. 158: I would add "scenarios" after "SMHI"

L. 164: if I got it right, Delta P close to 1 mean no change. Is it correct?

L. 172: "extenT"

L. 211: what you have done is a proxy-basin sample test according to the well-known paper Klemes (1986). This is not done so often, I recommend citing this paper.

L. 220: "temperatureS"

L. 225: I would add a comma after "September"

L. 302: I also see a shift of the peak for the far future

L. 320: "snow-fre"

L. 323: using the future is a bit too categorical. There are some uncertainties in projections.

L. 360: any ideas about these other uses? I think this is of interest for the readers.

L. 428: Farinotti et al. (2012) is given twice.

L. 471: Viviroli et al. (2004) has been published, please update

L. 480: "SIMULATION1": what is this "1"?

Table 1: space or no space between "mm" and "H2O"? In the caption, I would place "SWAT parameters" just after "sensitive"

Fig. 1 and 2: scale and north direction are missing. I would skip "The Damma Glacier CZO" on top of Fig. 1.

Fig. 3 and others: months are not given in English ("Dez"). I would also lie to see each time in the caption the catchment of interest and the period.

Fig. 5: panel (a) is too small for the long period given; it hides potential serious mismatches between simulation and observations.

Fig. 6: is it 1981 as in the text or 1987? Is that an interannual mean? Please comment why SWAT underestimates low flows.

**References**:

Schaefli, B. and Gupta, H. V. (2007), Do Nash values have value?. Hydrol. Process., 21: 2075-2080. doi:10.1002/hyp.6825

---

## Referee Comment (RC2) · Anonymous Referee #2 · 22 Nov 2018

The paper entitled *Hydrological modelling and future runoff of the Damma Glacier CZO watershed using SWAT. Validation of the model in the greater area of the Göscheneralpsee, Switzerland* presents a practical application of the SWAT model in an alpine catchment under actual climate conditions as well as future scenarios. Additionally, the authors present a model validation at different spatial scales. The model results are compared and briefly discussed with output from PREVAH and Alpine3D.

[Figure]

**1 General remarks:**

Even though the paper is about important issues in hydrology (model complexity, impact of climate change), the current version has several flaws.

As pointed out by Guillaume Thirel, its main goal is not clearly stated. You state that SWAT "has rarely been used for high alpine areas" and imply to study the suitability of SWAT for such environment. This is not completely true, as SWAT has been widely used in mountainous regions during the last decade (see for example Rahman et al. 2013, references within and papers citing it). The authors should carefully streamline the main goal of the paper.

A second major problem is the lack of references or justifications throughout the text. You make strong statements without justifying them or explaining why you made that choice. Here are a few examples:

- The calibration and validation periods are both very short (line 181-183). Why have you chosen such a limited period?

- You estimate the glacier retreat during the last 90 years (line 63-64) without any reference. Where does it come from?

- Climate models (line 147-151): why have you chosen these 3 models out of the 10 available in CH2011? is there any reason?

- To the best of my knowledge, the CH2011 scenarios (based on the delta change method) were not suitable for assessing changes in extreme events. Based on which element, are you stating an increase in extreme events (Line 342-343)?

You are making strong assertions based on the Nash-Sutcliffe model efficiency throughout the paper (line 197-198, 250-251, 259, 268), but be careful, because this

indicator strongly depends on the hydrological regime (Schaefli and Gupta, 2007). In alpine basins where you have a strong annual cycle, a NSE coefficient of 0.49 is rather bad and not satisfactory as you state. When comparing averaged models results (Figure 6, line 284-292), based on which elements (objective/subjective) can you say that the performance of SWAT is comparable to PREVAH and Alpine3D? I personally do not agree based on the NSE coefficients you provided.

Some of the SWAT parameters seem to be scale-dependent (in time and space), which could partly explain the model performance deterioration. You should somehow discuss which parameters are the most sensitive in space (validation over the Göschneneralpsee) and in time (with regard to climate scenarios). In addition, you are using different soil and landuse maps in the Damma and Göschneneralpsee catchments (Line 114-122). For me, this choice is a bit risky as you upscale your parameters and could bring some inconsistency.

**2  Minor remarks:**

Some typos are visible throughout the paper, the authors should carefully proofread it. Here are some minor comments:

1. Line 44: what do you mean by "its structure is physically based"? For me, Alpine3D is a physically based model, SWAT is not. Please clarify!

2. Line 98: what do you mean exactly by this statement?

3. Line 104: "basic input" is a subjective statement.

4. Line 124: the new MeteoSwiss network is named SwissMetNet not ANETZ anymore.

5. Line 1341-134: you are right, lapse rate are critical in mountainous regions, so tell the reader which values you have used in you study!

6. In figure 1, what is the added value of the inset for the present study? There is an inconsistency in the orientation (North) between figure 1 and 2. You should just combine them into a single figure.

7. Figure 3a, is it really useful to show the uncalibrated time series?

8. We can hardly see the difference between the two curves in figure 5a. Consequently, the reader cannot really assess the quality of the model.

9. In figure 6, it is somehow hard to make the difference between the lines. Try different colors.

---

## Editor Comment (EC1) · B. Schaefli (Editor) · 22 Nov 2018

Reviewer G. Thirel discusses that it is unclear what this paper contributes in terms of original research and states that "if the main additional value of the paper is the fact that SWAT works for this area, then this should be better highlighted and put into perspective with relevant literature".

I invite the authors to carefully answer this point. HESS does, in fact, not publish simple case studies, except so-called "cutting-edge-case studies" (see https://www.hydrology-

and-earth-system-sciences.net/about/manuscript_types.html), which come with the obligation of publishing the used data:

"Cutting-edge case studies report on case studies that require (a) broadening the knowledge base in hydrology as well as (b) sharing the underlying data and models. These case studies should be cutting edge with respect to the quality and diversity of data provided, the soundness of the models employed, and the importance of the study objective. Peer reviews of a manuscript of this type will additionally evaluate whether this case study points out challenges for hydrological science and/or accurate/appropriate/innovative design and execution of challenging experiments. The data and models used for the article have to be made publicly available through an open data repository or as a supplement. (..)"

---

## Editor Comment (EC2) · B. Schaefli (Editor) · 22 Nov 2018

The first reviewer discusses that the Nash criterion is not a good indicator for strongly seasonal signals. In this context, it seems questionable whether SWAT has any predictive power for the validation catchment (without re-calibration). The authors state: "The efficiency of inflow predictions (NS) dropped to 0.49 and the R2 to 0.72, which are however satisfactory. The observed and predictive accumulative flow is presented in Fig.5(b)".

[Figure]

A Nash value of 0.49 for such a strongly seasonal signal might have no predictive power (Schaefli and Gupta, 2007). A simple experiment illustrates this: if you generate a sine curve that has the same seasonality as the observed discharge, similar amplitude and the same mean, and no negative values (e.g. by shifting the sine curve), then the Nash value of this signal (compared to the observed discharge) most likely has a Nash value of between 0.4 and 0.5. Attached to this comment is a Matlab example, including an illustration.

Given the above, I think that we need more evidence that the model actually has predictive power. A key question is hereby whether the model can predict winter low flows (i.e. it gets the baseflow right), general timing of snow melt, general timing if high flows, autumn recession etc.

Please also note the supplement to this comment:
https://www.hydrol-earth-syst-sci-discuss.net/hess-2018-493/hess-2018-493-EC2-supplement.zip

———————————————

---

## Author Comment (AC1) · 8 Feb 2019

Dear Reviewer G. Thirel, thank you for your review and constructive comments. I hope that we answer all your remarks.

Reviewer: "The paper by Andrianaki et al. deals with a topic of interest for HESS readers: the modelling of runoff in a glacierised catchments and projections of its evolution. The manuscript reads easily and is concise; I would like to thank the authors for that, as it is often not the case and readers are burdened with loads of not so useful

information in many papers. That said, I feel that there is room for improvement before the paper reads as a scientific paper. Here are my main remarks. 1) The main criticism is that I failed to identify clearly what the readers could bring home from this manuscript. Definitely not a new methodology, as the SWAT model is basically used as is, the sensitivity test is not detailed and the calibration and climate change exercises are classical. In my opinion, results are also not so remarkable. It is very interesting to see the validation exercise on a different period and then on a different catchment, but in the end we have results about one catchment and the calibration period is very short. As aconsequence, we could wonder if we have the right answer for the right reason or not. I find it very difficult to extrapolate anything from results on this catchment for further works. If the main additional value of the paper is the fact that SWAT works for this area, then this should be better highlighted and put into perspective with relevant literature. This reflects on the objectives of the study, which are barely presented in the paper and makes it look like an application of the model rather than an actual research work. Only lines 51-52 give some elements on the interest of this work. Consistently, the conclusions only briefly highlight one novelty of the study (L. 354). In my opinion, the abstract, the introduction and the conclusions should be clear about the novelty of this work."

Authors: You are right that probably we didn't explain clearly enough the objectives of this study.

One of the challenges that researchers in hydrological modelling face is the lack of data for model setup and calibration in ungauged watersheds. Especially in high mountainous regions a big part of the watersheds is ungauged. In the last years, there is an increasing interest in applying SWAT on snow-dominated (Grousson et al., 2015) and glacierised watersheds (Omani et al., 2017; Rahman et al., 2013). However, its transponsability and its application for the simulation of runoff in high altitude ungauged watersheds hasn't been tested yet.

Our study area is characterised by extreme climatic conditions, high altitude and steep

slopes. Here, we have a quite unique situation; a small well gauged watershed monitored through the CZO projects, which is part of a larger watershed, for which we have hydrological data thanks to its use by the hydroelectric power plant. This gives us the opportunity to verify the applicability of SWAT under extreme conditions and its transferability on spatial and temporal scale, by using the small Damma watershed (10.5 km2) as the gauged watershed and the greater area feeding the Göscheneralpsee reservoir (100 km2) as an ungauged catchment. We used the approach of spatial proximity and transferred the calibrated parameters of the model from the donor watershed, which in this case is the Damma glacier watershed, to the greater area. By comparing the model results with the existing measurements provided by the managers of the reservoir, we were able to test whether the model can eventually be transposed and applied efficiently on a different spatial scale, and where its advantages and disadvantages lie.

Finally, we conducted the climate change simulations, not to do another set of classical climate change exercises, but to investigate whether SWAT can be further transposed on a temporal scale, since we could compare our findings with those of a previous study for the same area, which used two other hydrological models with different characteristics, PREVAH and Alpine 3D (Kobierska et al., 2013).

In addition, the Damma Glacier watershed is a Critical Zone Observatory part of the Critical Zone Exploration Network, a global network of field sites investigating the physical, chemical and biological processes of the critical zone (www.czen.org). Because CZOs are well studied sites and usually have long records of data, we wanted to show how they can be used in water management, since they could serve as parameter donor catchments.

Our results presented in the manuscript, as well as further analysis suggested by the Editor (please see our response to the Editor), showed that SWAT can predict satisfactorily runoff after being upscaled and applied in different scales, even under alpine conditions. This approach, which doesn't require complex regionalisation methods, can be quite useful in water management and climate change studies, considering the

fact that SWAT is a widely used model, even in large scale simulations (Pagliero et al, 2014). The performance of the model could be further improved if different rates of glaciermelt and snowmelt had been applied.

In the revised manuscript we have rewritten a big part of the abstract and introduction, adding relevant literature, discussing all the above with more detail and explaining the objectives in a clear way. In the conclusions paragraph we discussed in a more critical and constructive way about the performance of the model and how it could be improved and the conclusions from the comparison between the models and the climate change study.

Reviewer:"2) It is, if I'm not wrong, never clearly stated that calibration of SWAT is done compared todischarge observations only. Calibration is mentioned many times (abstract, end of introduction, section 3.3) but the used observation is not given. SWAT is physically based and snow observations are definitely an additional value to models calibration in snowy areas, so it is legitimate to wonder if the authors used any kind of snow data here."

Authors: The model was calibrated against measured runoff of the Damma watershed, which is described in paragraph 3.2.4. Comparison of the measured runoff with the results of the model before and after the calibration is given in Fig.3, page 16. Small corrections were made in the text to make this clearer.

Data for the evolution of the glacier were available for the whole area (paragraph 3.2.5) provided by Paul et al., 2007 and snow density and snow depth measurements were available for the Damma watershed only. We used these data to define the initial glacier storage for each elevation band of each subbasin. We didn't use it for the calibration of the model because we didn't think it would add to the purpose of the study at this stage.

Reviewer: "3) The calibration set up is unclear and at some point, flawed to me. First, we don't know exactly what the objective function is: authors introduce NS and $R^2$ but

they don't specify how they used them: through a composite criterion? With a Pareto front? Then, the use of NS in snowfed basins is not advised. Indeed, this criterion relates the performance to the mean observed discharge, which is a bad predictor in such a seasonally variable environment (see Schaefli and Gupta (2007)). It also underestimates discharge variability. Finally, we don't know how the parameters from the small basin are transferred to the larger one. Are some of these parameters time or scale dependent? It is just said that they are adjusted. 4) The structure of section 4.1 is not easy to follow. Some kind of sensitivity test is done to identify which parameters to calibrate. I failed to understand if it was done by the authors, and if yes I don't understand why it is mentioned only in the third paragraph, so after talking about the values of the calibrated parameters. Also, the word "set" is often used to refer to parameters; as it is unclear what is meant since both a manual calibration and an automatic one are mentioned, I got a bit lost. In addition, authors seem to infer that Table 1 shows the results of a sensitivity test. What I rather see here is how different the calibrated values are from the default ones, some of them being unrealistic maybe (I don't know where they come from). L. 239: which ones are the least sensitive ones?"

Authors: Initially we conducted the calibration manually because we wanted to identify the parameters that really influence the hydrology of the site. For the manual calibration both NS and R2 were checked but again manually. After the manual calibration we used SWAT-CUP software and the program SUFI-2 (Sequential Uncertainty Fitting version 2) (Abbaspour et al., 2007) for the automated calibration (fine tuning) and the sensitivity analysis. The manual calibration helped us in defining which parameters will be calibrated by SUFI-2 as well as their range. For example, because our site is above the tree line, evapotranspiration is not significant, and ET related parameters were left to their default values. For the SUFI-2 NS objective function was chosen because it was the criterion available in SUFI-2, which is most commonly used in similar studies.

Table 1 doesn't show the sensitivity test. It shows the default and calibrated values of the parameters that were introduced into SUFI2 and were calibrated. The sensitivity

test showed that these parameters are indeed the most sensitive ones. Some of these values are very different from the default ones probably because our watershed is characterised by extreme conditions. For example, due to its topography (very steep slopes) and geology Damma watershed has a very fast response which led to the high value of ALPHA_BF and the low value of GW_DELAY.

The input data of SWAT include topography, landuse and soil maps and during the initial delineation of the watershed many parameters are given a value based on these data. This a priori parameterisation assisted the use of the model for the bigger area. Then the calibrated parameters were applied to the bigger area with the same values that resulted from the calibration without any regionalisation procedure or another adjustment. We decided to do that because the Damma watershed and the greater area are very similar.

After receiving your review, we calculated the Benchmark Efficiency according to Schaefli and Gupta (2007) and for the period 2009-2011 the BE value is 0.22 and for 2012-2015 the BE is 0.25. The calculation of BE is included now in the revised text. Furthermore, more detail was added in the calibration paragraph to make it better understood.

Reviewer:"5) The actual setup of this whole study is not justified by the authors. Why is the modelcalibrated on the small basin that has few data and validated on the large basin with a lot ofdata rather than the opposite?"

Authors: As mentioned above, in this study we have a quite unique situation; a small well gauged watershed monitored through the CZO projects, which is part of a larger watershed, for which we have hydrological data thanks to its use by the hydroelectric power plant. This way we wanted to check the application of SWAT in high altitude basins and its upscaling to ungauged catchments in alpine conditions. Since we already had the climate change study with Alpine 3D and PREVAH for the bigger area, we calibrated the model for the small watershed and transferred it to the bigger. In the

revised text we give more detail to explain this further.

Reviewer: "6) L. 304: I thought that the black (reference) curve in Fig. 7 should be the same as the SWATcurve in Fig. 6, but it does not seem so. Did I get something wrong? The resolution of Fig. 7could be improved, it is more difficult to read than Fig. 6."

Authors: You are right. There is an error in the text, line 284. In Fig. 6 the interannual average is for the period 1997-2010 and not 1981–2010 mentioned in the text. The caption of Fig.6 is correct. In Fig. 7 the reference period is 1981-2010. Figures 6 and Figures 7 were redone.

Reviewer: "7) L. 317: the authors state that the volume of the glacier reduces to half in 2070. I wonder how this is considered in the SWAT model. Indeed, I expect that the initialconditions of the model (due to the Delta method used for producing the climate projections a continuoushydrological projection cannot be done) had to be adjusted. How was that done? Also,please precise who estimated this reduction (authors? Literature?)."

Authors: We have data for the evolution of the glacier for both future periods provided by Paul et al. (2007). Based on this, the initial glacier storage was calculated, and the SWAT was setup for each climate change scenario. According to the data of the evolution of the glaciers the glacier volume will be reduced in our site approximately to half by 2070. The sentence was rephrased to explain this better.

Minor remarks:

Reviewer: "Title: The title is not very sexy... Also CZO is an acronym, is it well known enough to be used in a title?" Authors: Indeed, the title is not very sexy. Another title could be "Assessment of the transferability of SWAT at an alpineglacierised catchment". CZO is removed from the title anyway.

Reviewer:"L. 30, 32 and many other places: a space is missing after the semi-colon." Authors: Corrected

Reviewer: "L. 31: I think that the lack of observed data of sufficient quality could also be mentioned." Authors: Done

Reviewer: "Section 2: what is the surface area of the small watershed? It is only given for the larger one." Authors: The area of the small watershed is 10.5 km2

Reviewer: "L. 60: after "(Fig. 1)" I think that "is" is missing." Authors: Done

Reviewer: "L. 62: inconsistent (lack of) space between number and unit." Authors: Corrected

Reviewer:"L. 69, 74. . .: why is "et al." suddenly in italics?" Authors: Corrected

Reviewer: "L. 77: I would add a comma after "interface"" Authors: Corrected

Reviewer:"L. 135: strange punctuation after "Climate change scenarios" Authors: Corrected

Reviewer:L. 149-150: are the parentheses necessary around Delta P and Delta T? "(Bosshard et al. 2011)" should be "Bosshard et al. (2011)" Authors: Corrected

Reviewer:L. 158: I would add "scenarios" after "SMHI" Authors: Added

Reviewer:L. 164: if I got it right, Delta P close to 1 mean no change. Is it correct? Authors: Yes

Reviewer:L. 172: "extenT" Authors: Corrected

Reviewer:L. 211: what you have done is a proxy-basin sample test according to the well-known paper Klemes (1986). This is not done so often, I recommend citing this paper Authors: You are right that our approach is similar to the proxy basin sample test suggested by Klemes (1986) and we added this paper in the introduction together with a short description of the test.

Reviewer:L. 220: "temperatureS" Authors: Corrected

Reviewer:L. 225: I would add a comma after "September" Authors: Corrected

Reviewer:L. 302: I also see a shift of the peak for the far future Authors: The sentence in L.302 was deleted because it was not clear enough.

Reviewer:L. 320: "snow-fre" Authors: Corrected

Reviewer:L. 323: using the future is a bit too categorical. There are some uncertainties in projections. Authors: Some sentences were rephrased to emphasize that these are predictions.

Reviewer:L. 360: any ideas about these other uses? I think this is of interest for the readers. Authors: This approach could be used in the simulation of runoff in high altitude ungauged catchments with limited data or in large scale simulations with SWAT. Big part of the paragraph was rephrased to explain this in a better way.

Reviewer: L. 428: Farinotti et al. (2012) is given twice. L. 471: Viviroli et al. (2004) has been published, please update L. 480: "SIMULATION1": what is this "1"? Authors: Corrected

Reviewer:Table 1: space or no space between "mm" and "H2O"? In the caption, I would place "SWAT parameters" just after "sensitive" Authors: Corrected

Reviewer: Fig. 1 and 2: scale and north direction are missing. I would skip "The Damma Glacier CZO" on top of Fig. 1.

Authors: Figures 1 and 2 were combined in one.

Reviewer: Fig. 3 and others: months are not given in English ("Dez"). I would also lie to see each time in the caption the catchment of interest and the period. Authors: Sorry for not noticing about the months that are not in English. It is corrected. The captions were corrected to include catchment and period.

Reviewer:Fig. 5: panel (a) is too small for the long period given; it hides potential serious mismatches between simulation and observations. Authors: We tried to apply a different colour scheme and it is slightly improved.
Reviewer: Fig. 6: is it 1981 as in the text or 1987? Is that an interannual mean? Please comment why SWAT underestimates low flows. Authors: The caption is correct. The 1981 in the text was wrong but now is corrected. It is true that SWAT underestimates low flows and this discussion is added in the revised manuscript. The Damma glacier watershed is characterised by very steep slopes (even up to nearly 80 degrees) and runoff originates mainly from snowmelt, glacier melt and rainfall (Magnuson et al., 2012). Consequently, the watershed is characterised by very fast response, which in terms of the model parameters resulted on the high value of ALPHA_BF and the low value of the GW_Delay. On the other hand, the Göscheneralpsee feeding area is less steep on average and maybe the interactions between groundwater and surface runoff must be more significant than those of the Damma watershed. Furthermore, two out of the four watersheds of the greater area are drained into the reservoir through tunnels, which undoubtably influence the low flow measurements of the reservoir. These factors explain why the model, which is calibrated for the Damma watershed, doesn't simulate successfully the low flows of the greater area.

References

Abbaspour, K. C., Yang, J., Maximov, I., Siber, R., Bogner, K., Mieleitner, J., Zobrist, J., and Srinivasan, R.: Modelling hydrology and water quality in the pre-alpine/alpine Thur watershed using SWAT, J. Hydrol., 333, 413-430, http://dx.doi.org/10.1016/j.jhydrol.2006.09.014, 2007. Grusson, Y., Sun, X., Gascoin, S., Sauvage, S., Raghavan, S., Anctil, F., and Sáchez-Pérez, J.-M.: Assessing the capability of the SWAT model to simulate snow, snow melt and streamflow dynamics over an alpine watershed, J. Hydrol., 531, 574-588, http://dx.doi.org/10.1016/j.jhydrol.2015.10.070, 2015. Omani, N., Srinivasan, R., Karthikeyan, R., and Smith, P.: Hydrological Modeling of Highly Glacierized Basins (Andes, Alps, and Central Asia), Water, 9, 111, 2017. Paul, F., Maisch, M., Rothenbühler, C., Hoelzle, M., and Haeberli, W.: Calculation and visualisation of future glacier extent in the Swiss Alps by means of hypsographic modelling, Global

and Planetary Change, 55, 343-357, https://doi.org/10.1016/j.gloplacha.2006.08.003, 2007. Pagliero, L., Bouraoui, F., Willems, P., and Diels, J.: Large-Scale Hydrological Simulations Using the Soil Water Assessment Tool, Protocol Development,and Application in the Danube Basin, Journal of Environmental Quality, 43, 145–154, http://dx.doi:10.2134/jeq2011.0359, 2014. Rahman, K., Maringanti, C., Beniston, M., Widmer, F., Abbaspour, K., and Lehmann, A.: Streamflow Modeling in a Highly Managed Mountainous Glacier Watershed Using SWAT: The Upper Rhone River Watershed Case in Switzerland, Water Resour. Manag., 27, 323-339, 10.1007/s11269-012-0188-9, 2013.

www.czen.org
* * *

---

## Author Comment (AC2) · 8 Feb 2019

Dear Editor we thank you for your comment.

Editor: The first reviewer discusses that the Nash criterion is not a good indicator for strongly seasonal signals. In this context, it seems questionable whether SWAT has any predictive power for the validation catchment (without re-calibration). The authors state: "The efficiency of inflow predictions (NS) dropped to 0.49 and the R2 to 0.72, which are however satisfactory. The observed and predictive accumulative flow is presented in Fig.5(b)".

A Nash value of 0.49 for such a strongly seasonal signal might have no predictive power (Schaefli and Gupta, 2007). A simple experiment illustrates this: if you generate a sine curve that has the same seasonality as the observed discharge, similar amplitude and the same mean, and no negative values (e.g. by shifting the sine curve), then the Nash value of this signal (compared to the observed discharge) most likely has a Nash value of between 0.4 and 0.5. Attached to this comment is a Matlab example, including an illustration.

Given the above, I think that we need more evidence that the model actually has predictive power. A key question is hereby whether the model can predict winter low flows (i.e. it gets the baseflow right), general timing of snow melt, general timing if high flows, autumn recession etc.

In response to your comment, we investigated further whether the model can predict the spring snowmelt timing, timing of highest flow, autumn recession period and the centre of mass (COM). We used the 15-day average of the daily runoff and results are presented in Fig. 1 and 2 and the Table 1 given below. Figure 1 shows the observed and simulated spring snowmelt timing and Fig. 2 the highest flow timing for each year of the period 1997-2010. Table 1 shows the difference in days between the observed and simulated centre of mass and autumn recession period.

The model predicts satisfactorily the spring snowmelt timing and the autumn recession period. The difference between the COM of the observed and the simulated runoff, Table 1, is low and for some years close to zero, which is also satisfactory. Overall, we believe that these additional data prove the predictability of the model.

There are only some inconsistencies between measured and simulated data for the general timing of the highest flow, Fig. 2, and for certain years. These inconsistencies have two possible explanations. Firstly, the Damma glacier watershed is characterised by very steep slopes (even up to nearly 80 degrees) and runoff originates mainly from snowmelt, glacier melt and rainfall (Magnuson et al., 2012). Consequently the watershed is characterised by very fast response, which in terms of the model parameters resulted on the high value of ALPHA_BF and the low value of the GW_Delay. On the other hand, the Göscheneralpsee

feeding area is less steep on average and for the two out of the four of its watersheds, runoff is drained through tunnels into the reservoir. These two factors explain the difference in the response and the fact that the simulated runoff peaks are higher and narrower than the observed ones.

In addition, the model doesn't differentiate between snow and glacier dynamics and only one parameter for both snowmelt and a glaciermelt rate is applied. This becomes more important in our study, since there is a difference between the glacier coverage of the two catchments. The Damma glacier is 50% covered by the glacier while the greater catchment is 20%.

Table 1 Difference of the centre of mass (COM) and autumn recession period in days, calculated from the 15-day average.

| Year | COM | Autumn recession period |
|------|-----|-------------------------|
| 1997 | 6.8 | 1 |
| 1998 | 4.2 | 1 |
| 1999 | 1.0 | 0 |
| 2000 | 3.0 | 16 |
| 2001 | 0.6 | 1 |
| 2002 | 7.8 | 19 |
| 2003 | 0.6 | 5 |
| 2004 | 2.4 | 4 |
| 2005 | 4.3 | 0 |
| 2006 | 4.1 | 1 |
| 2007 | 8.1 | 1 |
| 2008 | 3.1 | 0 |
| 2009 | 4.6 | 0 |
| 2010 | 6.0 | 0 |

[Figure]

*Figure 1 Comparison between the observed and simulated spring snowmelt timing. A 15-day average filter was applied on daily measurements.*

[Figure]

Figure 2 Comparison between the observed and simulated spring snowmelt timing. A 15-day average filter was applied on daily measurements.

---

## Author Comment (AC3) · 8 Feb 2019

Dear Anonymous referee, thank you for your review and very constructive comments.

**General remarks**

Reviewer: Even though the paper is about important issues in hydrology (model complexity, impact of climate change), the current version has several flaws. As pointed out by Guillaume Thirel, its main goal is not clearly stated. You state that SWAT "has rarely been used for high alpine areas" and imply to study the suitability of SWAT for such environment. This is not completely true, as SWAT has been widely used in mountainous regions during the last decade (see for example Rahman et al. 2013, references within and papers citing it). The authors should carefully streamline the main goal of the paper.

Authors: You are right that the main goal of the paper is not clear. In this manuscript, we wanted to show not only the applicability of SWAT on a glacierised watershed but also to assess its transferability in different spatial and temporal scale and subsequently to test whether it can be applied on a high altitude glacierised ungauged watershed for runoff simulation and climate change simulations. This is something that hasn't been done before with SWAT but can be quite useful in water management considering the fact that SWAT is a widely used model, used even in large scale simulations (Pagliero et al, 2014).

It is true that in the last years there is an increasing interest in the application of SWAT in high mountainous areas and since a big part of the watersheds in these regions is ungauged, we believe that our study can contribute towards this direction. In our site we have the opportunity to test the upscaling of the model, because we have a quite unique situation; a small well gauged watershed monitored through the CZO projects, which is part of a larger watershed and for which we have hydrological data thanks to its use forthe hydroelectric power plant. This gives us the opportunity to verify the model with independently collected data on the large watershed.

We have rewritten the abstract and conclusions, and extended the introduction focusing on the points mentioned above in order to make our objectives clearer. We also added relevant literature to put them into perspective.

Reviewer: A second major problem is the lack of references or justifications throughout the text. You make strong statements without justifying them or explaining why you made that choice. Here are a few examples:
• The calibration and validation periods are both very short (line 181-183). Why have you chosen such a limited period?

Authors: The reason why the calibration and validation periods are short is that for the Damma glacier watershed we had runoff data for the period 2009-2013. Probably it would be best if we had used these runoff data only for calibration and omitted the validation step, but

the performance of the calibration is the same as the calibration and therefore we think that it wouldn't make any difference. In addition, this period is short, but it still includes a relatively large variability in the weather conditions and precipitation amounts. For example, it includes a rather wet year and hot summer and dry and warm autumn.

Reviewer: You estimate the glacier retreat during the last 90 years (line 63-64) without any reference. Where does it come from?

Authors: Damma glacier watershed is a well studied site. Glacier retreat was estimated in previous studies described in Bernasconi et al., 2008 and Bernasconi et al., 2011 (already cited in the paragraph) using systematic recordings.

Reviewer: • Climate models (line 147-151): why have you chosen these 3 models out of the 10 available in CH2011? is there any reason?

Authors: We used these three models, because they were theones in common with both Alpine 3D and PREVAH.

Reviewer: To the best of my knowledge, the CH2011 scenarios (based on the delta change method) were not suitable for assessing changes in extreme events. Based on which element, are you stating an increase in extreme events (Line 342-343)?

Authors: What we meant is that predicted runoff of the far future period T2 shows higher fluctuations from year to year than that of the near future period especially from September to October. Sentence is rephrased.

Reviewer: You are making strong assertions based on the Nash-Sutcliffe model efficiency throughout the paper (line 197-198, 250-251, 259, 268), but be careful, because this indicator strongly depends on the hydrological regime (Schaefli and Gupta, 2007). In alpine basins where you have a strong annual cycle, a NSE coefficient of 0.49 is rather bad and not satisfactory as you state. When comparing averaged models results (Figure 6, line 284-292), based on which elements (objective/subjective) can you say that the performance of SWAT is comparable to PREVAH and Alpine3D? I personally do not agree based on the NSE coefficients you provided.

Some of the SWAT parameters seem to be scale-dependent (in time and space), which could partly explain the model performance deterioration. You should somehow discuss which parameters are the most sensitive in space (validation over the Göschneneralpsee) and in time (with regard to climate scenarios). In addition, you are using different soil and landuse maps in the Damma and Göschneneralpsee catchments (Line114-122). For me, this choice is a bit risky as you upscale your parameters and could bring some inconsistency

Authors: In response to your comment and the comment by the Editor, we investigated further the predictive power of the model for the greater catchment by comparing the observed data with the model results for the spring snowmelt timing, timing of highest flow, autumn recession period and the centre of mass (COM). To do this analysis we used the 15-day average of the daily runoff. Results are presented in Fig. 1 and 2 and the Table 1 given below.

The model predicts efficiently the spring snowmelt timing and the autumn recession period. The difference between the COM of the observed and the simulated runoff, which is given in Table 1, is low and for some years close to zero, which is also satisfactory. The main inconsistencies between measured and simulated data are observed for the general timing of the highest peak, Fig. 2.

One of the reasons for the deterioration of the model is that it doesn't differentiate between snow and glacier dynamics and only one parameter for both snowmelt and glacier meltrates is applied. This becomes more important in our study, since there is a difference between the glacier coverage of the two catchments. The Damma glacier is 50% covered by the glacier while the greater catchment is 20%.

One more reason is the difference in the response of the Damma glacier watershed in comparison to the greater area. Damma is characterised by very steep slopes (even up to nearly 80 degrees) and runoff originates mainly from snowmelt, glacier melt and rainfall (Magnuson et al., 2012). For this reason, the small watershed is characterised by very fast response, which led to the high value of ALPHA_BF and the low value of the GW_Delay parameters. On the other hand, the Göscheneralpsee feeding area is less steep on average and for the two out of the four of its watersheds, runoff is drained through tunnels into the reservoir.

The most sensitive parameters are the ones related to the snowmelt, like SFTMP, SMTMP and TIMP. During the manual calibration we checked many of the parameters related to landuse and soils and we think that we do not have an inconsistency. The parameter values set during the delineation of the watershed and initial parameterisation should be adequate. Finally, because our site is above tree line evapotranspiration parameters are not significant.

It is true that comparing SWAT with Alpine3D and PREVAH is tricky since they were calibrated for different catchments and different periods of time. The NS efficiency and the benchmark efficiency BE (added in the revised text)for the calibration period only are: 0.85 and 0.19 respectively for ALPINE3D, 0.91 and 0.49 for PREVAH and 0.84 and 0.22 for SWAT. These efficiencies of Alpine 3D and SWAT are in good agreement, with the efficiencies of PREVAH being slightly higher.

We have rewritten the entire paragraph for the comparison of the models. We focused less on comparing the efficiency of the model and mainly on what we can conclude from the comparison between the three models.

**Minor remarks**

Reviewer: Some typos are visible throughout the paper, the authors should carefullyproofread it.Here are some minor comments:

Authors: We will proofread the paper

Reviewer:1. Line 44: what do you mean by "its structure is physically based"? For me, Alpine3D is a physically based model, SWAT is not. Please clarify!

Authors: It really depends on how you define the term "physically based". Some researchers consider SWAT to be a physically based model and others don't since not all of its parameters can be defined directly by measurements. Since it wasn't adding to the context, the sentence was deleted.

Reviewer:2. Line 98: what do you mean exactly by this statement?

Authors: Fontaine et al., (2002) revealed the importance of improving SWAT algorithms to include in the model the influence of elevation and season on the dynamics of the snowpack.

Reviewer: 3. Line 104: "basic input" is a subjective statement.

Authors: "basic" is deleted

Reviewer: Line 124: the new MeteoSwiss network is named SwissMetNet not ANETZ anymore.

Authors: Corrected

Reviewer: 5. Line 1341-134: you are right, lapse rate are critical in mountainous regions, so tell the reader which values you have used in you study!

Authors: precipitation lapse rate PLAPS was set to 5 (mm/km) and temperature lapse rate was set to -5.84 ($^\circ$C/km).

Reviewer: 6. In figure 1, what is the added value of the inset for the present study? There is aninconsistency in the orientation (North) between figure 1 and 2. You should justcombine them into a single figure.

Authors: You are right. The Figures will be combined.

Reviewer: 7. Figure 3a, is it really useful to show the uncalibrated time series?

Authors: We wanted to show that SWAT cannot be used here without calibration.

Reviewer: 8. We can hardly see the difference between the two curves in figure 5a. Consequently, the reader cannot really assess the quality of the model

Authors: A better version of Fig. 5a is given below in Fig. 3. As you can see in this Fig. as well as in Fig. 5b in the manuscript, that shows the accumulative runoff, there is an overestimation of the streamflow by the model during the years 2000 to 2002. This overestimation must be related to the runoff melt rate but we need to investigate this further by looking into the weather data. Furthermore, the simulated runoff peaks are higher and narrower than the observed ones, which must be related to the differences in the response and groundwater interactions between the small watershed and the greater area, as discussed above.

Reviewer: 9. In figure 6, it is somehow hard to make the difference between the lines. Try different colors.

Authors: Is Fig. 4 here clear enough?

**References**

Bernasconi, S. M., Christl, I., Hajdas, I., Zimmermann, S., Hagedorn, F., Smittenberg, R. H., Furrer, G., Zeyer, J., Brunner, I., Frey, B., Plotze, M., Lapanje, A., Edwards, P., Venterink, H. O., Goransson, H., Frossard, E., Bunemann, E., Jansa, J., Tamburini, F., Welc, M., Mitchell, E., Bourdon, B., Kretzschmar, R., Reynolds, B., Lemarchand, E., Wiederhold, J., Tipper, E., Kiczka, M., Hindshaw, R., Stahli, M., Jonas, T., Magnusson, J., Bauder, A., Farinotti, D., Huss, M., Wacker, L., Abbaspour, K., and Biglink Project, M.: Weathering, soil formation and initial ecosystem evolution on a glacier forefield: a case study from the Damma Glacier, Switzerland, Mineral. Mag., 72, 19-22, 10.1180/minmag.2008.072.1.19, 2008.

Bernasconi, S. M., Bauder, A., Bourdon, B., Brunner, I., Bunemann, E., Christl, I., Derungs, N., Edwards, P., Farinotti, D., Frey, B., Frossard, E., Furrer, G., Gierga, M., Goransson, H., Gulland, K., Hagedorn, F., Hajdas, I., Hindshaw, R., Ivy-Ochs, S., Jansa, J., Jonas, T., Kiczka, M., Kretzschmar, R., Lemarchand, E., Luster, J., Magnusson, J., Mitchell, E. A. D., Venterink, H. O., Plotze, M., Reynolds, B.,

Smittenberg, R. H., Stahli, M., Tamburini, F., Tipper, E. T., Wacker, L., Welc, M., Wiederhold, J. G., Zeyer, J., Zimmermann, S., and Zumsteg, A.: Chemical and Biological Gradients along the Damma Glacier Soil Chronosequence, Switzerland, Vadose Zone J., 10, 867-883,http://dx.doi:10.2136/vzj2010.0129, 2011.

Kobierska, F., Jonas, T., Zappa, M., Bavay, M., Magnusson, J., and Bernasconi, S. M.: Future runoff from a partly glacierized watershed in Central Switzerland: A two-model approach, Advances in Water Resources, 55, 204-214, http://dx.doi.org/10.1016/j.advwatres.2012.07.024, 2013.

Pagliero, L., Bouraoui, F., Willems, P., and Diels, J.: Large-Scale Hydrological Simulations Using the Soil Water Assessment Tool, Protocol Development,and Application in the Danube Basin, Journal of Environmental Quality, 43, 145–154, http://dx.doi:10.2134/jeq2011.0359, 2014.

Table 1 Difference of the centre of mass (COM) and autumn recession period in days, calculated from the 15-day average.

| Year | COM | Autumn recessionperiod |
|------|-----|------------------------|
| 1997 | 6.8 | 1 |
| 1998 | 4.2 | 1 |
| 1999 | 1.0 | 0 |
| 2000 | 3.0 | 16 |
| 2001 | 0.6 | 1 |
| 2002 | 7.8 | 19 |
| 2003 | 0.6 | 5 |
| 2004 | 2.4 | 4 |
| 2005 | 4.3 | 0 |
| 2006 | 4.1 | 1 |
| 2007 | 8.1 | 1 |
| 2008 | 3.1 | 0 |
| 2009 | 4.6 | 0 |
| 2010 | 6.0 | 0 |

[Figure]

*Figure 1 Comparison between the observed and simulated spring snowmelt timing. A 15-day average filter was applied on daily measurements.*

[Figure]

Figure 2 Comparison between the observed and simulated spring snowmelt timing. A 15-day average filter was applied on daily measurements.

[Figure]

Figure 3(Figure 5a in manuscript) SWAT results and measured runoff values of the feeding catchment of the Göscheneralpsee for the period 1997-2010

[Figure]

Figure 4 (Figure 6 in manuscript) Interannual mean of the results of SWAT, Alpine3D and PREVAH models and the measured runoff of the Göscheneralpsee feeding catchment for the 1997-2010 period.

---

## Author Comment (AC5) · 8 Feb 2019

Dear Editor,

our study is not just a simple case study. We tried to answer this point carefully in the responses to the reviewers and we revised our manuscript in order to make our objectives clearer.

Best regards

---

## Editor Comment (EC3) · Bettina Schaefli (Editor) · 13 Feb 2019

Reviewer 1 asks about the use of snow data for model calibration. The authors do notfurther answer why they did not use snow data (in-situ or remotely-sensed).In exchange, they discuss that there are glacier observations but simply state at the end of their response: "We didn't use it for the calibration of the model because we didn'tthink it would add to the purpose of the study at this stage."

This response is surprising since hydrological model development in high alpine areas is known to strongly benefit from snow and glaciers observations. Such data in particular is required to assess whether the model gives the right answer for the right reasons.In my view, not using any additional data for model calibration is not acceptable for apaper whose purpose is to show the potential of the model in this kind of environments(see the scope of the paper in the response to reviewer 2: "In this manuscript, wewanted to show not only the applicability of SWAT on a glacierised watershed but alsoto assess its transferability in different spatial and temporal scale and subsequently totest whether it can be applied on a high altitude glacierised ungauged watershed forrunoff simulation and climate change simulations).

If the authors maintain that additional data is not useful for the purpose of this study,this should be carefully justified in the revised version.

---

## Editor Comment (EC4) · Bettina Schaefli (Editor) · 13 Feb 2019

Both reviewers come to the conclusion that the paper is of potential interest for the readers of HESS but that it requires major revisions and re-review before it can be published. These revisions apply to various aspects of the paper and the underlying model analyses. Above all, the objective of the paper and how this adds to existing literature has to be much clearer. Important methodological choices have to be better explained and referenced. I invite the authors to submit a revised version that carefully addresses all the raised points along the lines of the public discussion. I would like

to underline that this revision and corresponding "answers to reviewers comments" (rebuttal) should be as concise and explicit as possible in terms of methodological choices.

---

## Author Comment (AC6) · 1 Apr 2019

Dear Editor,

We would like to thank you and the reviewers for your very constructive comments, that led to a significant improvement of this manuscript. Here we discuss the revisions that we made in response to your reviews and we hope that we addressed all the comments and issues raised by the reviewers.

Firstly, in response to the comment from reviewer 1, G. Thirel, that the title is not very fitting, we suggest a change in the title, if this is possible. The suggested new title is: "Assessment of SWAT spatial and temporal transferability for high altitude glacierised catchments". In addition, we updated in the manuscript one of the coauthors` affiliation. The structure of the manuscript was modified in order to address the main issue raised by the reviewers and concerns the goal of this study and to clarify methodological choices. Two new sections were added; one called Methodology and the second Model setup, calibration and validation. The abstract, introduction and conclusions were completely revised as well as other sections of the text. Figures 1 and 2 as well as 3 and 4 were combined in two new ones, that better describe the study site and the calibration data. A Table was also added with the new analysis data.

Below you will find our answers to the reviewers. A manuscript with all the changes is submitted together with this letter. Please note that the line and Figures numbers have been modified and that we refer to the new numbers in the responses below.

Yours sincerely

Maria Andrianaki (on behalf of all authors)

Reviewer 1

Dear Reviewer G. Thirel, thank you for your review and constructive comments. I hope that we answer all your remarks.

Reviewer: "The paper by Andrianaki et al. deals with a topic of interest for HESS readers: the modelling of runoff in a glacierised catchments and projections of its evolution. The manuscript reads easily and is concise; I would like to thank the authors for that, as it is often not the case and readers are burdened with loads of not so useful information in many papers. That said, I feel that there is room for improvement before the paper reads as a scientific paper. Here are my **main remarks**.

1) The main criticism is that I failed to identify clearly what the readers could bring home from this manuscript. Definitely not a new methodology, as the SWAT model is basically used as is, the sensitivity test is not detailed and the calibration and climate change exercises are classical. In my opinion, results are also not so remarkable. It is very interesting to see the validation exercise on a different period and then on a different catchment, but in the end we have results about one catchment and the calibration period is very short. As aconsequence, we could wonder if we have the right answer for the right reason or not. I find it very difficult to extrapolate anything from results on this catchment for further works.

If the main additional value of the paper is the fact that SWAT works for this area, then this should be better highlighted and put into perspective with relevant literature. This reflects on the objectives of the study, which are barely presented in the paper and makes it look like an application of the model rather than an actual research work. Only lines 51-52 give some elements on the interest of this work. Consistently, the conclusions only briefly highlight one novelty of the study (L. 354). In my opinion, the abstract, the introduction and the conclusions should be clear about the novelty of this work."

Authors: You are right that probably we didn't explain clearly enough the objectives of this study.

One of the challenges that researchers in hydrological modelling face is the lack of data for model setup and calibration in ungauged watersheds. Especially in high mountainous regions a big part of the watersheds is ungauged. In the last years, there is an increasing interest in applying SWAT on snow-dominated (Grousson et al., 2015) and glacierised watersheds (Omani et al., 2017; Rahman et al., 2013). However, its transponsability and its application for the simulation of runoff in high altitude ungauged watersheds hasn't been tested yet.

Our study area is characterised by extreme climatic conditions, high altitude and steep slopes. Here, we have a quite unique situation; a small well gauged watershed monitored through the CZO projects, which is part of a larger watershed, for which we have hydrological data thanks to its use by the hydroelectric power plant. This gives us the opportunity to verify the

applicability of SWAT under extreme conditions and its transferability on spatial and temporal scale, by using the small Damma watershed (10.5 km$^2$) as the gauged watershed and the greater area feeding the Göscheneralpsee reservoir (100 km$^2$) as an ungauged catchment. We used the approach of spatial proximity and transferred the calibrated parameters of the model from the donor watershed, which in this case is the Damma glacier watershed, to the greater area. By comparing the model results with the existing measurements provided by the managers of the reservoir, we were able to test whether the model can eventually be transposed and applied efficiently on a different spatial scale, and where its advantages and disadvantages lie.

Finally, we conducted the climate change simulations, not to do another set of classical climate change exercises, but to investigate whether SWAT can be further transposed on a temporal scale, since we could compare our findings with those of a previous study for the same area, which used two other hydrological models with different characteristics, PREVAH and Alpine 3D (Kobierska et al., 2013).

In addition, the Damma Glacier watershed is a Critical Zone Observatory part of the Critical Zone Exploration Network, a global network of field sites investigating the physical, chemical and biological processes of the critical zone (www.czen.org). Because CZOs are well studied sites and usually have long records of data, we wanted to show how they can be used in water management, since they could serve as parameter donor catchments.

Our results presented in the manuscript, as well as further analysis suggested by the Editor (please see our response to the Editor), showed that SWAT can predict satisfactorily runoff after being upscaled and applied in different scales, even under alpine conditions. This approach, which doesn't require complex regionalisation methods, can be quite useful in water management and climate change studies, considering the fact that SWAT is a widely used model, even in large scale simulations (Pagliero et al, 2014). The performance of the model could be further improved if different rates of glacier melt and snowmelt had been applied.

In the revised manuscript we have rewritten a big part of the abstract and introduction, adding relevant literature, discussing all the above with more detail and explaining the objectives in a clear way. In the conclusions paragraph we discussed in a more critical and constructive way about the performance of the model and how it could be improved and the conclusions from the comparison between the models and the climate change study.

Reviewer:"2) It is, if I'm not wrong, never clearly stated that calibration of SWAT is done compared todischarge observations only. Calibration is mentioned many times (abstract, end of introduction, section 3.3) but the used observation is not given. SWAT is physically based and snow observations are definitely an additional value to models calibration in snowy areas, so it is legitimate to wonder if the authors used any kind of snow data here."

Authors: The model was calibrated against measured runoff of the Damma watershed, which is described in paragraph 3.2.4. Comparison of the measured runoff with the results of the model after the calibration is given in Fig. 2 (former Fig.3), page 19. Small corrections were made in the text to make this clearer.

Data for the evolution of the glacier were available for the whole area (paragraph 3.2.5) provided by Paul et al., 2007 and snow density and snow depth measurements were available for the Damma watershed only. We used the evolution of the glacier to define the initial glacier storage for each elevation band of each subbasin. We didn't use it for the calibration of the model because we wanted to test the performance of the model following a simpler approach that would be familiar to the majority of SWAT users and that would also relate to studies with data scarcity where snow measurements are not available. We also think that the best way to improve SWAT performance in this case would be to take into account the difference between snowmelt and glacier melt dynamics. Omani et al. (2017) addressed this issue by applying different snow parameters to the completely glacierised subbasins. However, the subbasins of the Damma glacier watershed were all partly glacierised so we couldn't follow this approach.

Reviewer: "3) The calibration set up is unclear and at some point, flawed to me. First, we don't know exactly what the objective function is: authors introduce NS and $R^2$ but they don't specify how they used them: through a composite criterion? With a Pareto front? Then, the use of NS in snowfed basins is not advised. Indeed, this criterion relates the performance to the mean observed discharge, which is a bad predictor in such a seasonally variable environment (see Schaefli and Gupta (2007)). It also underestimates discharge variability.
Finally, we don't know how the parameters from the small basin are transferred to the larger one. Are some of these parameters time or scale dependent? It is just said that they are adjusted.
4) The structure of section 4.1 is not easy to follow. Some kind of sensitivity test is done to identify which parameters to calibrate. I failed to understand if it was done by the authors, and if yes I don't understand why it is mentioned only in the third paragraph, so after talking about the values of the calibrated parameters. Also, the word "set" is often used to refer to parameters; as it is unclear what is meant since both a manual calibration and an automatic one are mentioned, I got a bit lost.
In addition, authors seem to infer that Table 1 shows the results of a sensitivity test. What I rather see here is how different the calibrated values are from the default ones, some of them being unrealistic maybe (I don't know where they come from). L. 239: which ones are the least sensitive ones?"

Authors: Initially we conducted the calibration manually because we wanted to identify the parameters that really influence the hydrology of the site. For the manual calibration both NS and $R^2$ were checked but again manually. After the manual calibration we used SWAT-CUP

software and the program SUFI-2 (Sequential Uncertainty Fitting version 2) (Abbaspour et al., 2007) for the automated calibration (fine tuning) and the sensitivity analysis. The manual calibration helped us in defining which parameters will be calibrated by SUFI-2 as well as their range. For example, because our site is above the tree line,evapotranspiration is not significant, and ET related parameters were left to their default values. For the SUFI-2 NS objective function was chosen because it was the criterion available in SUFI-2, which is most commonly used in similar studies.

Table 1 doesn't show the sensitivity test. It shows the default and calibrated values of the parameters that were introduced into SUFI2 and were calibrated. The sensitivity test showed that these parameters are indeed the most sensitive ones. Some of these values are very different from the default ones probably because our watershed is characterised by extreme conditions. For example, due to its topography (very steep slopes) and geology Damma watershed has a very fast response which led to the high value of ALPHA_BF and the low value of GW_DELAY.

The input data of SWAT include topography, landuse and soil maps and during the initial delineation of the watershed many parameters are given a value based on these data. This a priori parameterisation assisted the use of the model for the bigger area. Then the calibrated parameters were applied to the bigger area with the same values that resulted from the calibration without any regionalisation procedure or another adjustment. We decided to do that because the Damma watershed and the greater area are very similar.

After receiving your review, we calculated the Benchmark Efficiency according to Schaefli and Gupta (2007) and for the period 2009-2011 the BE value is 0.22 and for 2012-2015 the BE is 0.25. The calculation of BE is included now in the revised text. Furthermore, more detail was added in the calibration paragraph to make it better understood. The calibration, criteria, sensitivity analysis and results are presented altogether in section 5.

Reviewer:"5) The actual setup of this whole study is not justified by the authors. Why is the modelcalibrated on the small basin that has few data and validated on the large basin with a lot ofdata rather than the opposite?"

Authors: As mentioned above, in this study we have a quite unique situation; a small well gauged watershed monitored through the CZO projects, which is part of a larger watershed, for which we have hydrological data thanks to its use by the hydroelectric power plant. This way we wanted to check the application of SWAT in high altitude basins and its upscaling to ungauged catchments in alpine conditions. Since we already had the climate change study with Alpine 3D and PREVAH for the bigger area, we calibrated the model for the small watershed and transferred it to the bigger. In the revised text we explain this further by adding more detail in the Abstract, Introduction and conclusion as well as in the added section 4 Methodology.

Reviewer: "6) L. 304: I thought that the black (reference) curve in Fig. 7 should be the same as the SWATcurve in Fig. 6, but it does not seem so. Did I get something wrong? The resolution of Fig. 7could be improved, it is more difficult to read than Fig. 6."

Authors: You are right. There is an error in the text, (former line 284) now in line 366. In Figure 4 (former Fig. 6) the interannual average is for the period 1997-2010 and not 1981–2010 mentioned in the text. The caption of Fig.6 is correct. In Fig. 5 (former Fig.7) the reference period is 1981-2010. Former Figures 6 and Figures 7 were redone and are now Fig. 4 and 5.

Reviewer: "7) L. 317: the authors state that the volume of the glacier reduces to half in 2070. I wonder how this is considered in the SWAT model. Indeed, I expect that the initialconditions of the model (due to the Delta method used for producing the climate projections a continuoushydrological projection cannot be done) had to be adjusted. How was that done? Also,please precise who estimated this reduction (authors? Literature?)."

Authors: Line 402 – 403. We have data for the evolution of the glacier for both future periods provided by Paul et al. (2007). Based on this, the initial glacier storage was calculated, and the SWAT was setup for each climate change scenario. According to the data of the evolution of the glaciers the glacier volume will be reduced in our site approximately to half by 2070. The sentence was rephrased to explain this better.

Minor remarks:

Reviewer: "Title: The title is not very sexy… Also CZO is an acronym, is it well known enough to be used in a title?"
Authors: Indeed, the title is not very sexy. We suggest another title could be "Assessment of SWAT spatial and temporal transferability for high altitude glacierised catchments". CZO is removed from the title.

Reviewer:"L. 30, 32 and many other places: a space is missing after the semi-colon."
Authors: Corrected

Reviewer: "L. 31: I think that the lack of observed data of sufficient quality could also be mentioned."
Authors: Done. Currently L. 38

Reviewer: "Section 2: what is the surface area of the small watershed? It is only given for the larger one."
Authors: The area of the small watershed is 10.5 km$^2$. Now it is added in the text. L. 75

Reviewer: "L. 60: after "(Fig. 1)" I think that "is" is missing."

Authors: Done, Figure 1 was merged with Figure 2

Reviewer: "L. 62: inconsistent (lack of) space between number and unit."
Authors: Corrected

Reviewer:"L. 69, 74…: why is "et al." suddenly in italics?"
Authors: Corrected

Reviewer: "L. 77: I would add a comma after "interface"""
Authors: Corrected

Reviewer:"L. 135: strange punctuation after "Climate change scenarios"
Authors: Corrected, L.150

Reviewer: L. 149-150: are the parentheses necessary around Delta P and Delta T? "(Bosshard et al. 2011)" should be "Bosshard et al. (2011)"
Authors: Corrected, L. 155

Reviewer:L. 158: I would add "scenarios" after "SMHI"
Authors: Added, L. 165

Reviewer:L. 164: if I got it right, Delta P close to 1 mean no change. Is it correct?
Authors: Yes

Reviewer:L. 172: "extenT"
Authors: Corrected, L. 187

Reviewer: L. 211: what you have done is a proxy-basin sample test according to the well-known paper Klemes (1986). This is not done so often, I recommend citing this paper
Authors: You are right that our approach is similar to the proxy basin sample test suggested by Klemes (1986) and we added this paper in the introduction together with a short description of the test in section 4.

Reviewer:L. 220: "temperatureS"
Authors: Corrected, L. 241

Reviewer:L. 225: I would add a comma after "September"
Authors: Corrected

Reviewer:L. 302: I also see a shift of the peak for the far future
Authors: The sentence in L.302 was deleted because it was not clear enough.

Reviewer: L. 320: "snow-fre"

Authors: Corrected

Reviewer: L. 323: using the future is a bit too categorical. There are some uncertainties in projections.

Authors: Some sentences were rephrased to emphasize that these are predictions. L.382 - 435

Reviewer: L. 360: any ideas about these other uses? I think this is of interest for the readers.

Authors: This approach could be used in the simulation of runoff in high altitude ungauged catchments with limited data or in large scale simulations with SWAT. Big part of the conclusions, now section 7, was rephrased to explain this in a better way.

Reviewer: L. 428: Farinotti et al. (2012) is given twice. L. 471: Viviroli et al. (2004) has been published, please update L. 480: "SIMULATION1": what is this "1"?

Authors: Errors in the references were corrected

Reviewer: Table 1: space or no space between "mm" and "H2O"? In the caption, I would place "SWAT parameters" just after "sensitive"

Authors: Corrected

Reviewer: Fig. 1 and 2: scale and north direction are missing. I would skip "The Damma Glacier CZO" on top of Fig. 1.

Authors: Figures 1 and 2 were combined in one. L. 644

Reviewer: Fig. 3 and others: months are not given in English ("Dez"). I would also lie to see each time in the caption the catchment of interest and the period.

Authors: Former Figures 3 and 4 were combined in 1 and is now Figure 2. Sorry for not noticing about the months that are not in English. It is corrected. The captions were corrected to include catchment and period. The graph with the results from the default parameters was deleted

Reviewer: Fig. 5: panel (a) is too small for the long period given; it hides potential serious mismatches between simulation and observations.

Authors: Figure 3 (former Fig. 5) We tried to apply a different colour scheme and it is slightly improved.

Reviewer: Fig. 6: is it 1981 as in the text or 1987? Is that an interannual mean? Please comment why SWAT underestimates low flows.

Authors: Figure 4 (former Fig. 6) The caption is correct. The 1981 in the text was wrong but now is corrected. It is true that SWAT underestimates low flows and this discussion is added

in the revised manuscript L. 344 - 359. The Damma glacier watershed is characterised by very steep slopes (even up to nearly 80 degrees) and runoff originates mainly from snowmelt, glacier melt and rainfall (Magnuson et al., 2012). Consequently, the watershed is characterised by very fast response, which in terms of the model parameters resulted on the high value of ALPHA_BF and the low value of the GW_Delay. On the other hand, the Göscheneralpsee feeding area is less steep on average and maybe the interactions between groundwater and surface runoff must be more significant than those of the Damma watershed. Furthermore, two out of the four watersheds of the greater area are drained into the reservoir through tunnels, which undoubtably influence the low flow measurements of the reservoir. These factors explain why the model, which is calibrated for the Damma watershed, doesn't simulate successfully the low flows of the greater area.

**References**

Abbaspour, K. C., Yang, J., Maximov, I., Siber, R., Bogner, K., Mieleitner, J., Zobrist, J., and Srinivasan, R.: Modelling hydrology and water quality in the pre-alpine/alpine Thur watershed using SWAT, J. Hydrol., 333, 413-430, http://dx.doi.org/10.1016/j.jhydrol.2006.09.014, 2007.

Grusson, Y., Sun, X., Gascoin, S., Sauvage, S., Raghavan, S., Anctil, F., and Sáchez-Pérez, J.-M.: Assessing the capability of the SWAT model to simulate snow, snow melt and streamflow dynamics over an alpine watershed, J. Hydrol., 531, 574-588, http://dx.doi.org/10.1016/j.jhydrol.2015.10.070, 2015.

Omani, N., Srinivasan, R., Karthikeyan, R., and Smith, P.: Hydrological Modeling of Highly Glacierized Basins (Andes, Alps, and Central Asia), Water, 9, 111, 2017.

Paul, F., Maisch, M., Rothenbühler, C., Hoelzle, M., and Haeberli, W.: Calculation and visualisation of future glacier extent in the Swiss Alps by means of hypsographic modelling, Global and Planetary Change, 55, 343-357, https://doi.org/10.1016/j.gloplacha.2006.08.003, 2007.

Pagliero, L., Bouraoui, F., Willems, P., and Diels, J.: Large-Scale Hydrological Simulations Using the Soil Water Assessment Tool, Protocol Development, and Application in the Danube Basin, Journal of Environmental Quality, 43, 145–154, http://dx.doi:10.2134/jeq2011.0359, 2014.

Rahman, K., Maringanti, C., Beniston, M., Widmer, F., Abbaspour, K., and Lehmann, A.: Streamflow Modeling in a Highly Managed Mountainous Glacier Watershed Using SWAT: The Upper Rhone River Watershed Case in Switzerland, Water Resour. Manag., 27, 323-339, 10.1007/s11269-012-0188-9, 2013.

www.czen.org

Dear Anonymous referee, thank you for your review and very constructive comments.

**General remarks**

Reviewer: Even though the paper is about important issues in hydrology (model complexity, impact of climate change), the current version has several flaws. As pointed out by Guillaume Thirel, its main goal is not clearly stated. You state that SWAT "has rarely been used for high alpine areas" and imply to study the suitability of SWAT for such environment. This is not completely true, as SWAT has been widely used in mountainous regions during the last decade (see for example Rahman et al. 2013, references within and papers citing it). The authors should carefully streamline the main goal of the paper.

Authors: You are right that the main goal of the paper is not clear. In this manuscript, we wanted to show not only the applicability of SWAT on a glacierised watershed but also to assess its transferability in different spatial and temporal scale and subsequently to test whether it can be applied on a high altitude glacierised ungauged watershed for runoff simulation and climate change simulations. This is something that hasn't been done before with SWAT but can be quite useful in water management considering the fact that SWAT is a widely used model, used even in large scale simulations (Pagliero et al, 2014).

It is true that in the last years there is an increasing interest in the application of SWAT in high mountainous areas and since a big part of the watersheds in these regions is ungauged, we believe that our study can contribute towards this direction. In our site we have the opportunity to test the upscaling of the model, because we have a quite unique situation; a small well gauged watershed monitored through the CZO projects, which is part of a larger watershed and for which we have hydrological data thanks to its use forthe hydroelectric power plant. This gives us the opportunity to verify the model with independently collected data on the large watershed.

We have rewritten the abstract and conclusions, and extended the introduction focusing on the points mentioned above in order to make our objectives clearer. We also added relevant literature to put them into perspective.

Reviewer: A second major problem is the lack of references or justifications throughout the text. You make strong statements without justifying them or explaining why you made that choice. Here are a few examples:
• The calibration and validation periods are both very short (line 181-183). Why have you chosen such a limited period?

Authors: The reason why the calibration and validation periods are short is that for the Damma glacier watershed we had runoff data for the period 2009-2013. Probably it would be best if we had used these runoff data only for calibration and omitted the validation step, but the performance of the calibration is the same as the calibration and therefore we think that it wouldn't make any difference. In addition, this period is short, but it still includes a relatively large variability in the weather conditions and precipitation amounts. For example, it includes a rather wet year and hot summer and dry and warm autumn.

Reviewer: You estimate the glacier retreat during the last 90 years (line 63-64) without any reference. Where does it come from?

Authors: Damma glacier watershed is a well studied site. Glacier retreat was estimated in previous studies described in Bernasconi et al., 2008 and Bernasconi et al., 2011 (already cited in the paragraph) using systematic recordings.

Reviewer: • Climate models (line 147-151): why have you chosen these 3 models out of the 10 available in CH2011? is there any reason?

Authors: We used these three models, because they were the ones in common with both ALPINE3D and PREVAH.

Reviewer: To the best of my knowledge, the CH2011 scenarios (based on the delta change method) were not suitable for assessing changes in extreme events. Based on which element, are you stating an increase in extreme events (Line 342-343)?

Authors: What we meant is that predicted runoff of the far future period T2 shows higher fluctuations from year to year than that of the near future period especially from September to October. Sentence is rephrased.

Reviewer: You are making strong assertions based on the Nash-Sutcliffe model efficiency throughout the paper (line 197-198, 250-251, 259, 268), but be careful, because this indicator strongly depends on the hydrological regime (Schaefli and Gupta, 2007). In alpine basins where you have a strong annual cycle, a NSE coefficient of 0.49 is rather bad and not satisfactory as you state. When comparing averaged models results (Figure 6, line 284-292), based on which elements (objective/subjective) can you say that the performance of SWAT is comparable to PREVAH and Alpine3D? I personally do not agree based on the NSE coefficients you provided.

Some of the SWAT parameters seem to be scale-dependent (in time and space), which could partly explain the model performance deterioration. You should somehow discuss which parameters are the most sensitive in space (validation over the Göschneneralpsee) and in time (with regard to climate scenarios). In addition, you are using different soil and landuse maps

in the Damma and Göschneneralpsee catchments (Line114-122). For me, this choice is a bit risky as you upscale your parameters and could bring some inconsistency

Authors: In response to your comment and the comment by the Editor, we investigated further the predictive power of the model for the greater catchment by comparing the observed data with the model results for the spring snowmelt timing, timing of highest flow, autumn recession period and the centre of mass (COM). To do this analysis we used the 15-day average of the daily runoff. Results are presented in Fig. 1 and 2 and the Table 1 given below.

The model predicts efficiently the spring snowmelt timing and the autumn recession period. The difference between the COM of the observed and the simulated runoff, which is given in Table 1 here (Table 2 in the manuscript), is low and for some years close to zero, which is also satisfactory. The main inconsistencies between measured and simulated data are observed for the general timing of the highest peak, Fig. 2 (Table 2 in the manuscript).

One of the reasons for the deterioration of the model is that it doesn't differentiate between snow and glacier dynamics and only one parameter for both snowmelt and glacier meltrates is applied. This becomes more important in our study, since there is a difference between the glacier coverage of the two catchments. The Damma glacier is 50% covered by the glacier while the greater catchment is 20%.

One more reason is the difference in the response of the Damma glacier watershed in comparison to the greater area. Damma is characterised by very steep slopes (even up to nearly 80 degrees) and runoff originates mainly from snowmelt, glacier melt and rainfall (Magnuson et al., 2012). For this reason, the small watershed is characterised by very fast response, which led to the high value of ALPHA_BF and the low value of the GW_Delay parameters. On the other hand, the Göscheneralpsee feeding area is less steep on average and for the two out of the four of its watersheds, runoff is drained through tunnels into the reservoir.

The most sensitive parameters are the ones related to the snowmelt, like SFTMP, SMTMP and TIMP. During the manual calibration we checked many of the parameters related to landuse and soils and we think that we do not have an inconsistency. The parameter values set during the delineation of the watershed and initial parameterisation should be adequate. Finally, because our site is above tree line evapotranspiration parameters are not significant.

It is true that comparing SWAT with Alpine3D and PREVAH is tricky since they were calibrated for different catchments and different periods of time. The NS efficiency and the benchmark efficiency BE (added in the revised text)for the calibration period only are: 0.85 and 0.19 respectively for ALPINE3D, 0.91 and 0.49 for PREVAH and 0.84 and 0.22 for SWAT. These

efficiencies of Alpine 3D and SWAT are in good agreement, with the efficiencies of PREVAH being slightly higher.

We have rewritten the entire paragraph for the comparison of the models. We focused less on comparing the efficiency of the model and mainly on what we can conclude from the comparison between the three models.

**Minor remarks**

Reviewer: Some typos are visible throughout the paper, the authors should carefullyproofread it.Here are some minor comments:

Authors: Corrected

Reviewer:1. Line 44: what do you mean by "its structure is physically based"? For me, Alpine3D is a physically based model, SWAT is not. Please clarify!

Authors: It really depends on how you define the term "physically based". Some researchers consider SWAT to be a physically based model and others don't since not all of its parameters can be defined directly by measurements. Since it wasn't adding to the context, the sentence was deleted.

Reviewer:2. Line 98: what do you mean exactly by this statement?

Authors: L. 115 (former L. 98) Fontaine et al., (2002) revealed the importance of improving SWAT algorithms to include in the model the influence of elevation and season on the dynamics of the snowpack.

Reviewer: 3. Line 104: "basic input" is a subjective statement.

Authors: Line 122 (former L. 104) "basic" is deleted

Reviewer: Line 124: the new MeteoSwiss network is named SwissMetNet not ANETZ anymore.

Authors: Corrected

Reviewer: 5. Line 1341-134: you are right, lapse rate are critical in mountainous regions, so tell the reader which values you have used in you study!

Authors: Line 148. precipitation lapse rate PLAPS was set to 5 (mm/km) and temperature lapse rate was set to -5.84 ($^{o}$C/km).

Reviewer: 6. In figure 1, what is the added value of the inset for the present study? There is aninconsistency in the orientation (North) between figure 1 and 2. You should justcombine them into a single figure.

Authors: You are right. Figures 1 and 2 were combined to Figure 1.

Reviewer: 7. Figure 3a, is it really useful to show the uncalibrated time series?

Authors: Figures 3 and 4 were combined to one, Figure 2 and the uncalibrated time series was not included.

Reviewer: 8. We can hardly see the difference between the two curves in figure 5a. Consequently, the reader cannot really assess the quality of the model

Authors: A better version of Fig. 5a is given below in Fig. 3. (Fig. 3 in the manuscript). As you can see this Figure, there is an overestimation of the streamflow by the model during the years 2000 to 2002. This overestimation must be related to the runoff melt rate that 1999-2002 was a rather wet period. Furthermore, the simulated runoff peaks are higher and narrower than the observed ones, which must be related to the differences in the response and groundwater interactions between the small watershed and the greater area, as discussed above.

Reviewer: 9. In figure 6, it is somehow hard to make the difference between the lines. Try different colors.

Authors: Figure 6 is Figure 4 in the revised manuscript and a different colour scheme was applied.

**References**

Bernasconi, S. M., Christl, I., Hajdas, I., Zimmermann, S., Hagedorn, F., Smittenberg, R. H., Furrer, G., Zeyer, J., Brunner, I., Frey, B., Plotze, M., Lapanje, A., Edwards, P., Venterink, H. O., Goransson, H., Frossard, E., Bunemann, E., Jansa, J., Tamburini, F., Welc, M., Mitchell, E., Bourdon, B., Kretzschmar, R., Reynolds, B., Lemarchand, E., Wiederhold, J., Tipper, E., Kiczka, M., Hindshaw, R., Stahli, M., Jonas, T., Magnusson, J., Bauder, A., Farinotti, D., Huss, M., Wacker, L., Abbaspour, K., and Biglink Project, M.: Weathering, soil formation and initial ecosystem evolution on a glacier forefield: a case study from the Damma Glacier, Switzerland, Mineral. Mag., 72, 19-22, 10.1180/minmag.2008.072.1.19, 2008.

Bernasconi, S. M., Bauder, A., Bourdon, B., Brunner, I., Bunemann, E., Christl, I., Derungs, N., Edwards, P., Farinotti, D., Frey, B., Frossard, E., Furrer, G., Gierga, M., Goransson, H., Gulland, K., Hagedorn, F., Hajdas, I., Hindshaw, R., Ivy-Ochs, S., Jansa, J., Jonas, T., Kiczka, M., Kretzschmar, R., Lemarchand, E., Luster, J., Magnusson, J., Mitchell, E. A. D., Venterink, H. O., Plotze, M., Reynolds, B., Smittenberg, R.

H., Stahli, M., Tamburini, F., Tipper, E. T., Wacker, L., Welc, M., Wiederhold, J. G., Zeyer, J., Zimmermann, S., and Zumsteg, A.: Chemical and Biological Gradients along the Damma Glacier Soil Chronosequence, Switzerland, Vadose Zone J., 10, 867-883,http://dx.doi:10.2136/vzj2010.0129, 2011.

Kobierska, F., Jonas, T., Zappa, M., Bavay, M., Magnusson, J., and Bernasconi, S. M.: Future runoff from a partly glacierized watershed in Central Switzerland: A two-model approach, Advances in Water Resources, 55, 204-214, http://dx.doi.org/10.1016/j.advwatres.2012.07.024, 2013.

Pagliero, L., Bouraoui, F., Willems, P., and Diels, J.: Large-Scale Hydrological Simulations Using the Soil Water Assessment Tool, Protocol Development,and Application in the Danube Basin, Journal of Environmental Quality, 43, 145–154, http://dx.doi:10.2134/jeq2011.0359, 2014.

Table 1 Difference of the centre of mass (COM) and autumn recession period in days, calculated from the 15-day average.

| Year | COM | Autumn recession period |
|------|-----|-------------------------|
| 1997 | 6.8 | 1 |
| 1998 | 4.2 | 1 |
| 1999 | 1.0 | 0 |
| 2000 | 3.0 | 16 |
| 2001 | 0.6 | 1 |
| 2002 | 7.8 | 19 |
| 2003 | 0.6 | 5 |
| 2004 | 2.4 | 4 |
| 2005 | 4.3 | 0 |
| 2006 | 4.1 | 1 |
| 2007 | 8.1 | 1 |
| 2008 | 3.1 | 0 |
| 2009 | 4.6 | 0 |
| 2010 | 6.0 | 0 |

[Figure]

*Figure 1 Comparison between the observed and simulated spring snowmelt timing. A 15-day average filter was applied on daily measurements.*

[Figure]

Figure 2 Comparison between the observed and simulated spring snowmelt timing. A 15-day average filter was applied on daily measurements.

[Figure]

Figure 3(Figure 3a in manuscript) SWAT results and measured runoff values of the feeding catchment of the Göscheneralpsee for the period 1997-2010

---

## Author Comment (AC7) · 1 Apr 2019

The comment was uploaded in the form of a supplement: https://www.hydrol-earth-syst-sci-discuss.net/hess-2018-493/hess-2018-493-AC7-supplement.pdf

---

## Author Comment (AC9) · 2 Apr 2019

Dear Editor,

many thanks for your comment on snow data. We are sorry that our response was not very clear. We used the glacier data for the set up of the model, to define the initial glacier coverage of our catchments, not only for the current situation but also for the climate change scenarios. What we probably did not make very clear in our response to the reviewer is that we also used observations of the glacier retreat and glacier melt

during the manual calibration of the model, to ensure that the total simulated water budget is correct. The reason why we didn't do something more is that we wanted to test the performance of the model following the most common approach of applying SWAT. We think that this would be useful in cases with scarcity of data, where snow measurements aren't available.

The best way to improve SWAT performance in this case is to take into account the difference between snowmelt and glacier melt dynamics. Omani et al. (2017) addressed this issue by applying different snow parameters to the completely glacierised subbasins and different to those that aren't glacier covered. However, the subbasins of the Damma glacier watershed are partly glacierised and for this reason we decided to apply only one set of snow parameters.

Best wishes

Reference: Omani, N., Srinivasan, R., Karthikeyan, R., and Smith, P.: Hydrological Modeling of Highly Glacierized Basins (Andes, Alps, and Central Asia), Water, 9, 111, 2017

---

## Referee Report (RR1)

**1    General remarks:**

The manuscript *Assessment of SWAT spatial and temporal transferability for high altitude glacierised catchments* has been significantly improved compared to the first version, especially the introduction and the methods section. Concerning the results, I am a bit annoyed because your comparisons with Alpine3D and Prevah became very qualitative (Lines 366-376, line 429-435) compared to the previous manuscript, but, at the same time, not really robust. In addition, your main argument to explain the lower performances of the model is the use of a single melt coefficient for snow and ice (Line 341-349). This choice is very questionable in an alpine basin with such a glacier coverage under current conditions. But with regards to the goal of the paper, this is a huge source of uncertainty that you add to your future runoff simulations. My question would be: how could you trust your predictions knowing that the importance of snow melt (which is not that well simulated by your set of parameters) will become higher while the influence of glacier melt will decrease?

In my opinion, this study is a good qualitative assessment of climate change impacts on an alpine basin but the quantitative aspect is very limited. Therefore I would recommend to the authors to mention it more explicitly in the manuscript.

**2    Specific comments:**

1. Line 32: what do you mean by "management induced environmental changes"?

2. Line 44: *large* instead of *great*

3. Figure 1: add a and b letters for each sub-part of the figure

4. Line 105: Please clarify what you mean by: *What is important in our study is that melted snow is handled by the model the same way as the water that comes from precipitation regarding the calculation of runoff and percolation*

5. Line 111-112: avoid repetitions (detailed, in detail)

6. Line 124: change the verb define.

7. Line 128-130: I am not convinced that you will *reduce the uncertainty of the calibration* by using *detailed soil and land use maps* even if it is a commendable effort. In the same paragraph, put the website reference in the bibliography.

8. Line 142-145: the meteorological parameters you enumerate are available at the Damma station but not all of them are available in Gütsch right?

9. Line 157-160: check the spelling of this sentence!!

10. Section 3.2.3: try to streamline a bit this section especially the part on climate change scenarios which is hard to read (repetitions, intermittent).

11. Line 192-194: the sentences are not really relevant for the reader.

12. Line 221: How can you have *two different but similar watersheds*? They have maybe similarities but they are not similar!

13. Line 235-250: these two paragraphs are hard to read. Try to stremaline them by putting the parameters name into bracket for example!

14. Line 279-289: If I understand well, the snow melt temperatures SMTMP is the threshold under which you have no melt. How do you justify a optimal value of 2.5C which is very high?

15. Line 296-299: Your statement is a bit confusing: About which "previous model" are you talking about? Morevover, you should remind to the reader that you are working with daily time steps. This strongly influences the NSE coefficient.

16. Line 310-311: I don't understand you argument about wet years. Why would your model be less skilled to simulate a wet year? You also mention this argument on line 325-326. Please clarify!

17. Line 317-318: this is not really new: SWAT has been used in glacierized basin in the past.

18. Figure 3a: you can hardly see anything on such graph.

19. Line 333: This is a good idea to evaluate the runoff timing. But why have you applied a 15-days moving average on you data? This is quite brute force and will necessarily smooth out the differences.

20. Line 351-359: as you have daily discharge observations, I am not convinced about the influence of the basin slope on the discharge response. This could have an impact at hourly time step.

---

## Author Response (AR2)

**Dear Editor,**

We would like to thank you and the reviewers for the second review of our manuscript, the great feedback and again the very constructive comments. Here we discuss the new revisions that we made and we hope that we addressed all the comments and issues raised by the reviewers.

The abstract and the sections 6.2 and 7 were updated in order to better explain the contribution of this study, further from being just a case study. The title was slightly changed again by making two small corrections and the new title is "Assessment of SWAT spatial and temporal transferability for a high altitude glacierised catchment". A new table, Table 3, was added.

Below you will find our responses to the reviewers. The line numbers have been modified and we refer to the new numbers in the revised manuscript (not the marked-up version). A manuscript with all the changes is submitted together with this letter.

Yours sincerely

Maria Andrianaki (on behalf of all authors)

Dear Guillaume Thirel,

Many thanks for your feedback. Here we hope that we will address all the new issues raised after your second review.

"This is my second review of this manuscript. I'm happy to see that the authors made substantial efforts to improve the document. Especially, we know can see a bit more easily what the interest of the study is. I however still raise points that need to be tackled. Please note that I am using the lines numbers of the modification-apparent version of the manuscript.

My main reserves about the paper concern the following two groups of lines: L. 382, 930-931, 1068-1070, 1092-1094 and related parts: this is impossible to deduce that from the fact that the discharge projections of the three models are rather similar! The three models could be all wrong! Usually, the most we can say about such results is that the confidence interval could be narrow. It the authors want to show that the model is transferable in time and climate conditions, then a calibration/evaluation exercise must be performed over sufficiently long and differing periods, such as the Klemes Differential SST or Thirel et al. (2015) introduce."

This issue was addressed in L 347-349, 392-404, 424-441 of the revised manuscript. The text was changed to focus mainly on the uncertainties of the climate change simulations and the fact that no further conclusions could be drawn other than a qualitative assessment.

Section 5.3 and following: I don't like the word « validation » here. Indeed, a model can be falsified, but not validated. At the maximum, it can be evaluated or assessed. I suggest rather using one of these proposed terms instead of validation/validated. This is even truer as due to the very short periods of time used, we only have an incomplete picture of the model behavior!

The term validation was changed to evaluation

**Minor remarks:**

Regarding the title, I suggest removing the plural to « catchments ». Indeed, only one catchment (incl. One subbasin) was used. In addition, this would be consistent with the abstract: « in a partly glacierised alpine catchment ». I therefore suggest the following: «Assessment of SWAT spatial and temporal transferability for a high-altitude glacierised catchment ».

Yes the word "catchments" could be turned into singular

L. 33: please consider replacing « i.e. » with « e.g. »: It was replaced in L30

L. 99: « to assesS »: corrected in L. 45

L. 122: please consider specifying what transferability you tested.: Specified in line 58

L. 238: no comma after « used »: Corrected in L.128

L. 296-299: please specify from which SRES scenario these numbers are taken.: The A1B scenario was added L.152  $\,$

L. 370: please consider replacing  $\ll$  feeding  $\gg$  with  $\ll$  that feeds  $\gg$ , in order to allow a better reading of the rest of the sentence.: Replaced in L.187

L. 372: I would add « input flow » before data.: Added in L.189

L. 390: « different but similar » is maybe not the best way to say what you mean: Corrected in L.207

L. 417: « was set to active in order »: I failed to understand that formulation: Corrected in L.225

L. 462: « y^is the... y- is the » EQ 1 and 2: please use the same notations! L252: the same notations were changed and the same ones were used for both equations

Section 5.2: I definitely don't see what the added value of this section is. In addition, I remain convinced that sensitivity analysis must be realized before the calibration. Former section 5.2 was indeed not adding to the context of the manuscript and was deleted.

L. 591: please replace the point after 0.86 with a space: Replaced in L286

L. 598: actually, there is a growing literature regarding the use of time-varying parameters n hydrology, so this is not completely true.: The sentence was rephrased L.292

L. 604: are you comparing the NS of the bigger basin with the one of the smallest one? This is quite not correct, as the benchmark that is used in NS, namely mean(Qobs), is different for the two basins. In addition, we don't know here what is the period of evaluation and the BE is not given.

L.302 - 304 We rephrased the paragraph to better explain what we meant and we added the BE. As BE is a far stricter criterion had the negative value of -1.

L. 624 and 928: « doesn't » -> does not: Corrected L. 906: please remove the comma after « reason »: Removed

Section 6.2: I guess that the reference in Fig. 5 is SWAT forced by observed climate (not actually observed discharge or SWAT forced by reference period climate scenarios) as climate scenarios come from a delta change approach. Please mention that if correct. Yes that is correct and it was mentioned in the text in L.355-356

L. 992-995: the future is still used here, although I mentioned in my previous review that this is definitely not advised in climate change impact studies. Due to the numerous and important sources of uncertainties in projections, it is impossible to state that the T increase will be 3.35°C! The use of conditional and of uncertainty bands would help being more thorough.

Future tense is no longer in section 6.2 and we were more thorough in the presenting the results.

Table 1: what is Na?: Na means not available but it was deleted from the table so that it does not cause confusion

Table 2: I would find actual values more informative that absolute values, as this could tell us whether the errors are systematic or not for example. The actual values were added and the average was deleted since it was not adding any value to the table

The actual values were added and the average was deleted since it was not adding any value to the table

Fig. 2: the y scale of panels c and D should have units in (mm) (same for following figure). In addition, the caption and especially its first part does not present accurately what is shown in the figure.: Figure 2 and its caption were corrected

Figure 4: 20150 -> 2050 : Corrected

Figure 5: please consider saying "the runoff simulated with SWAT" instead of "the simulated with SWAT runoff".:Corrected

**Dear reviewer,**

Many thanks for your feedback and again very constructive comments. Here we hope that we will address all the new issues raised after your second review.

**1 General remarks:**

The manuscript Assessment of SWAT spatial and temporal transferability for high altitude glacierised catchments has been significantly improved compared to the first version, especially the introduction and the methods section. Concerning the results, I am a bit annoyed because your comparisons with Alpine3D and Prevah became very qualitative (Lines 366-376, line 429-435) compared to the previous manuscript, but, at the same time, not really robust. In addition, your main argument to explain the lower performances of the model is the use of a single melt coefficient for snow and ice (Line 341-349). This choice is very questionable in an alpine basin with such a glacier coverage under current conditions. But with regards to the goal of the paper, this is a huge source of uncertainty that you add to your future runoff simulations. My question would be: how could you trust your predictions knowing that the importance of snowmelt (which is not that well simulated by your set of parameters) will become higher while the influence of glacier melt will decrease? In my opinion, this study is a good qualitative assessment of climate change impacts on an alpine basin but the quantitative aspect is very limited. Therefore, I would recommend to the authors to mention it more explicitly in the manuscript.

It is true that the application of the same snow melt parameters added a great source of uncertainty in the climate change simulations and it was mentioned more explicitly in the manuscript. This issue was addressed in the abstract and L 347-349, 392-404, 424-441 of the revised manuscript. The text was changed to focus mainly on the uncertainties of the climate change simulations and the fact that no further conclusions could be drawn other than a qualitative assessment.

A more thorough comparison of the three models would be very interesting, especially considering the fact that SWAT is not calibrated for the greater area. However, we thought that it would be better not to focus on this, since there is already the extensive comparison study of Alpine 2D and PREVAH by Kobierska et al. (2013).

**2 Specific comments:**

1. Line 32: what do you mean by "management induced environmental changes"? Since the sentence was confusing, it was deleted by the text

Line 44: large instead of great: Corrected
 Figure 1: add a and b letters for each sub-part of the figure: We deleted a and b from the text.

4. Line 105: Please clarify what you mean by: What is important in our study is that melted snow is handled by the model the same way as the water that comes from precipitation regarding the calculation of runoff and percolation

Since this sentence was not adding to the context, we deleted it from the manuscript.

5. Line 111-112: avoid repetitions (detailed, in detail): Corrected in L.1066. Line 124: change the verb define: Corrected in L.119

7. Line 128-130: I am not convinced that you will reduce the uncertainty of the calibration by using detailed soil and land use maps even if it is a commendable effort. In the same paragraph, put the website reference in the bibliography.

You are right that soil and landuse maps did not reduce the uncertainty and this is why the sentence was deleted.

The website was added in the references

8. Line 142-145: the meteorological parameters you enumerate are available at the Damma station but not all of them are available in Gütsch right?

The Gütsch meteorological parameters include the ones mentioned in the text apart from the incoming longwave radiation and the records are hourly. The text was corrected in L.128

9. Line 157-160: check the spelling of this sentence!! : The sentence was rephrased and moved to L150-152

10. Section 3.2.3: try to streamline a bit this section especially the part on climate change scenarios which is hard to read (repetitions, intermittent). The section was rephrased in L. 144-169

11. Line 192-194: the sentences are not really relevant for the reader. The sentences were deleted.

12. Line 221: How can you have two different but similar watersheds? They have maybe similarities but they are not similar! Corrected in L.207

13. Line 235-250: these two paragraphs are hard to read. Try to streamline them by putting the parameters name into bracket for example!

The paragraphs now in L. 221-234 were streamlined and are easier to read.

14. Line 279-289: If I understand well, the snow melt temperatures SMTMP is the threshold under which you have no melt. How do you justify an optimal value of 2.5C which is very high?

Yes, SMTMP is the temperature that the snowpack has to reach to start melting and it can take values up to 5°C.

2.5C agrees with the study of Omani et al., 2017 who applied SWAT for the Rhone river basin and the SMTMP was set at a range 0.5-3.0.

15. Line 296-299: Your statement is a bit confusing: About which "previous model" are you talking about? Moreover, you should remind to the reader that you are working with daily time steps. This strongly influences the NSE coefficient.

The paragraph was rephrased in L.279-282.

16. Line 310-311: I don't understand you argument about wet years. Why would your model be less skilled to simulate a wet year? You also mention this argument on line 325-326. Please clarify!

The reason why SWAT overestimates runoff when the precipitation is significantly higher is unclear. This is why

we left it in L. 299-300 only as an observation and removed it from other parts of the manuscript.

17. Line 317-318: this is not really new: SWAT has been used in glacierized basin in the past. The sentence was removed from the text.

**18. Figure 3a: you can hardly see anything on such graph.**

We only left it because it shows the overestimation of runoff during the years 1999-2002. But it could also be removed.

19. Line 333: This is a good idea to evaluate the runoff timing. But why have you applied a 15-days moving average on your data? This is quite brute force and will necessarily smooth out the differences.

Since we wanted to calculate snowmelt timing, we applied the 15-day average window to smooth out peaks coming from short term events. It facilitated in the calculation of the snowmelt timing without eliminating differences between the model and observed values, since we were still able to observe the inconsistencies in the highest peak.

**20. Line 351-359: as you have daily discharge observations, I am not convinced about the influence of the basin slope on the discharge response. This could have an impact at hourly time step.**

The values of ALPHA\_BF and GW\_DELAY parameters that describe adequately Damma watershed cannot fully describe the greater area. This is probably because Damma is a watershed with faster response and the groundwater surface interactions are less important, as it was found in previous studies. The paragraph in L. 321-329 was rephrased to better explain this.

[revised manuscript text omitted]

681Table 2 DifferenceAbsolute difference in days of the simulated from the
between simulated and observed valu measured
valueses of the snowmelt timing, autumn recession period, peak flow timing and the centre of mass (COM), for the
greater catchment feeding the Göscheneralpsee.

| 684
685 | Year    | Snowmelt
timing | Autumn
recession
period | Peak flow
timing | СОМ        |
|------------|---------|--------------------|-------------------------------|---------------------|------------|
| 686        | 1997    | 0                  | 1                             | - 48         | 7          |
| 687        | 1998    | 2                  | 1                             | - 2          | 4          |
| 688        | 1999    | - 4         | 0                             | - 27         | - 1 |
| 689        | 2000    | 0                  | - 16                   | 19                  | - 3 |
| 600        | 2001    | 0                  | 1                             | - 1          | 1          |
| 090        | 2002    | 0                  | - 19                   | 0                   | 8          |
| 691        | 2003    | 2                  | 5                             | - 2          | 1          |
| 692        | 2004    | 1                  | 4                             | 21                  | 2          |
| 693        | 2005    | 1                  | 0                             | - 1          | 4          |
| 694        | 2006    | 3                  | 1                             | - 3          | 4          |
| c05        | 2007    | 3                  | 1                             | - 7          | 8          |
| 095        | 2008    | - 2         | 0                             | 2                   | 3          |
| 696        | 2009    | 1                  | 0                             | 13                  | 5          |
| 697        | 2010    | 2                  | 0                             | - 1          | - 6 |
| 698        |         |                    |                               |                     |            |
| 699        | Average | <del>1.4</del>     | 3.5                           | <del>11.0</del>     | 4.0        |

703Table 3 Shift in days of the centre of mass (COM) and shift in the highest runoff peak of the interannual average704reservoir inflow for all the three scenarios. T1 and T2 stand for the T1 and T2 periods respectively.

| Parameter | Model  | ETHZ T1    | CNRM T1 | SHMI T1 | ETHZ T2    | CNRM T2 | SHMI T2 |
|------------------|---------------|------------|----------------|----------------|------------|----------------|----------------|
| COM shift        | SWAT   | -2  | -1      | 1       | -6  | -4      | 2       |
| (days)    | Alpine 3D     | -2  | -1      | 0       | -4  | -6      | 1       |
|                  | PREVAH | -6  | -2      | 3       | -7  | -8      | 3       |
|                  |               |            |                |                |            |                |                |
| Peak day shift   | SWAT   | -10 | 0       | 0       | -22 | -16     | -13     |
| (days)    | Alpine 3D     | -12 | -12     | -6      | -45 | -44     | -30     |
|                  | PREVAH | -29 | -16     | -6      | -43 | -39     | -38     |